# SCRAPL: Scattering Transform with Random Paths for Machine Learning

**Christopher Mitcheltree**[♭]   **Vincent Lostanlen**[♯]   **Emmanouil Benetos**[♭]   **Mathieu Lagrange**[♯]

[♭]Centre for Digital Music, Queen Mary University of London, UK
{c.mitcheltree, emmanouil.benetos}@qmul.ac.uk

[♯]Nantes Université, École Centrale Nantes, CNRS, LS2N, UMR 6004, France
{vincent.lostanlen, mathieu.lagrange}@ls2n.fr

## Abstract

The Euclidean distance between wavelet scattering transform coefficients (known as *paths*) provides informative gradients for perceptual quality assessment of deep inverse problems in computer vision, speech, and audio processing. However, these transforms are computationally expensive when employed as differentiable loss functions for stochastic gradient descent due to their numerous paths, which significantly limits their use in neural network training. Against this problem, we propose "**Sc**attering transform with **Ra**ndom **P**aths for machine **L**earning" (SCRAPL): a stochastic optimization scheme for efficient evaluation of multivariable scattering transforms. We implement SCRAPL for the joint time–frequency scattering transform (JTFS) which demodulates spectrotemporal patterns at multiple scales and rates, allowing a fine characterization of intermittent auditory textures. We apply SCRAPL to differentiable digital signal processing (DDSP), specifically, unsupervised sound matching of a granular synthesizer and the Roland TR-808 drum machine. We also propose an initialization heuristic based on importance sampling, which adapts SCRAPL to the perceptual content of the dataset, improving neural network convergence and evaluation performance. We make our code and audio samples available and provide SCRAPL as a Python package.

## 1 Introduction

A scattering transform (ST) is a wavelet-based nonlinear operator which decomposes a high-resolution input $x$ into a collection $\Phi x$ of low-resolution coefficients, known as *paths* (Mallat, 2012). Without loss of generality, let us consider a two-layer multivariable ST of a time-domain signal $x(t)$:

$$\Phi x(p,t,\lambda) = \rho\left(\left(\left|\,|\mathbf{W}x|\circledast\boldsymbol{\Psi}_p\right|\circledast\boldsymbol{\Psi}_0\right)(t,\lambda)\right). \tag{1}$$

In the equation above, $\mathbf{W}$ is a wavelet transform; the vertical bars denote complex modulus; the circled asterisk $\circledast$ denotes a multivariable convolution over time $t$ and wavelet scale $\lambda$; $\boldsymbol{\Psi}$ is a multivariable wavelet filterbank which is indexed by path $p$; $\boldsymbol{\Psi}_0$, i.e., $\boldsymbol{\Psi}_p$ with $p = 0$ is a multivariable low-pass filter; and $\rho$ is a pointwise nonlinearity, e.g., path normalization and logarithmic transformation.

The design of the filterbank $\boldsymbol{\Psi}$ aims at a tradeoff between three properties: invariance to rigid motion, stability to small deformations, and separation of sparse patterns (Mallat, 2016). In speech and audio processing, examples of such $\boldsymbol{\Psi}$ include "plain" time ST (Andén & Mallat, 2014), joint time–frequency scattering (JTFS) (Andén & Mallat, 2014), and spiral ST (Lostanlen & Mallat, 2016). In computer vision, examples include "plain" 2-D ST (Bruna & Mallat, 2013), joint roto-translation ST (Sifre & Mallat, 2013), and scalo-roto-translation ST (Oyallon et al., 2014).

The squared Euclidean distance between scattering coefficients, or *ST distance* for short, is:

$$d_\Phi(x,\tilde{x}) = \sum_{p=0}^{P-1}\sum_{t=0}^{T-1}\sum_{\lambda=0}^{\Lambda-1}\left|\Phi x(p,t,\lambda) - \Phi\tilde{x}(p,t,\lambda)\right|^2, \tag{2}$$

where $P$ is the number of paths; $T$ is the number of time samples; and $\Lambda$ is the number of scales. Behavioral studies suggest that ST distance is a good predictor of dissimilarity judgments between isolated sounds, for suitably chosen $\boldsymbol{\Psi}$ and $\rho$ (Patil et al., 2012; Lostanlen et al., 2021; Tian et al.,

2025). Relatedly, neurophysiology studies suggest that JTFS is a suitable idealized model of spectrotemporal receptive fields in the auditory cortex of humans (Norman-Haignere & McDermott, 2018) and nonhuman mammals (Kowalski et al., 1996). These findings motivate the use of JTFS as part of a differentiable loss function for neural audio models (Vahidi et al., 2023).

As an illustration, let $\boldsymbol{x}$ be a fixed reference and $\tilde{\boldsymbol{x}} = F_{\boldsymbol{x}}(\boldsymbol{w})$ be its reconstruction by an autoencoder $F$ with trainable weights $\boldsymbol{w}$. Denoting the set of path indices by $\mathscr{P} = \{0, \ldots, P-1\}$ and the vector of all time–frequency entries $\Phi\boldsymbol{x}(p, t, \lambda)$ for each path $p \in \mathscr{P}$ by $\phi_p(\boldsymbol{x})$, the ST loss function writes as:

$$\mathscr{L}_{\boldsymbol{x}}^{\Phi}(\tilde{\boldsymbol{x}}) = \frac{1}{P}\sum_{p=0}^{P-1}\mathscr{L}_{\boldsymbol{x}}^{\phi_p}(\tilde{\boldsymbol{x}}) \quad \text{where} \quad \forall p \in \mathscr{P},\ \mathscr{L}_{\boldsymbol{x}}^{\phi_p}(\tilde{\boldsymbol{x}}) = P\big\|\phi_p(\boldsymbol{x}) - \phi_p(\tilde{\boldsymbol{x}})\big\|^2. \tag{3}$$

Unfortunately, $\mathscr{L}_{\boldsymbol{x}}^{\Phi}(\tilde{\boldsymbol{x}})$ and its gradient $\nabla\mathscr{L}_{\boldsymbol{x}}^{\Phi}(\tilde{\boldsymbol{x}})$ are expensive in memory and in operations. Certainly, algorithmic refinements such as FFT-based filtering, multirate processing, and depth-first search can reduce the cost of an ST path (Oyallon et al., 2018). Yet, the need to traverse all $P$ paths remains an obstacle to the applicability of multivariable ST for gradient-based learning at scale.

In this article, we aim to accelerate the training of an autoencoder $F$ whose loss is the ST distance between reference and reconstruction, and so over a finite corpus $\mathscr{X} = \{\boldsymbol{x}_0, \ldots, \boldsymbol{x}_{N-1}\}$. Formally:

$$\boldsymbol{w}^{\star} = \arg\min_{\boldsymbol{w}} \frac{1}{N}\sum_{n=0}^{N-1}\left(\mathscr{L}_{\boldsymbol{x}_n}^{\Phi} \circ F_{\boldsymbol{x}_n}\right)(\boldsymbol{w}). \tag{4}$$

Given the decomposition in Equation 3, a naïve idea would be to replace each term $\mathscr{L}_{\boldsymbol{x}_n}^{\Phi}$ in the equation above by some per-path loss $\mathscr{L}_{\boldsymbol{x}_n}^{\phi_p}$, where the $p$'s would be drawn independently and uniformly at random in the path set $\mathscr{P}$. This is a crude form of stochastic approximation (Benveniste et al., 2012) which is motivated by the tree-like structure of ST: neglecting the overhead of the first layer ($|\mathbf{W}\boldsymbol{x}|$), the computation of a single-path gradient $\nabla\mathscr{L}_{\boldsymbol{x}_n}^{\phi_p}$ is roughly $P$ times more efficient than that of a full ST gradient $\nabla\mathscr{L}_{\boldsymbol{x}_n}^{\Phi}$. However, this speedup comes at the detriment of numerical precision: a deterministic quantity has been replaced by an estimator whose variance may be impractically large.

"**Sc**attering transform with **Ra**ndom **P**aths for machine **L**earning" (SCRAPL) is our proposed solution to this problem. Acknowledging that each single-path gradient makes for an inexpensive but noisy learning signal, we stabilize it via a combination of three stochastic optimization techniques and apply an architecture-informed importance sampling heuristic. Our contributions are:

1. **Stochastic approximation of scattering transform** through uniform sampling of paths.
2. **Path-wise adaptive moment estimation** ($\mathscr{P}$-Adam for short): an extension of the Adam algorithm (Kingma & Ba, 2014) which accounts for the non-i.i.d. nature of ST paths.
3. **Path-wise stochastic average gradient with acceleration** ($\mathscr{P}$-SAGA for short): a variant of the SAGA algorithm (Defazio et al., 2014) which keeps a memory of previous gradient values across all paths $p$.
4. **$\theta$-importance sampling**: a parallelizable initialization heuristic that supplies auxiliary information to the stochastic optimizer by sampling paths $p$ in proportion to the typical rate of change of the gradient in the optimization landscape.

Our main empirical finding is that SCRAPL accomplishes a favorable tradeoff between goodness of fit and computational efficiency on unsupervised sound matching, i.e., a nonlinear inverse problem in which the forward operator implements an audio synthesizer. In the context of differentiable digital signal processing (DDSP), the state-of-the-art perceptual loss function for this task is multiscale spectral loss (MSS, Yamamoto et al. (2020), Engel et al. (2020)). However, the gradient of MSS is uninformative when input and reconstruction are misaligned or when the synthesizer controls involve spectrotemporal modulations (Vahidi et al., 2023). Taking advantage of the stability guarantees of JTFS, SCRAPL expands the class of synthesizers which can be effectively decoded via DDSP.

Figure 1 illustrates one of our experiments: unsupervised sound matching for a nondeterministic granular synthesizer. On one hand, models based on MSS and other state-of-the-art perceptual losses are computationally efficient but inaccurate. On the other hand, JTFS-based models are five times more accurate but twenty five times more costly. SCRAPL is a new point on this Pareto front: it is within a factor two of JTFS in terms of accuracy while being within a factor two of MSS in terms of runtime, making it suitable for large-scale DDSP. Relatedly, SCRAPL is also more memory-efficient than JTFS, thus reducing overhead between CPU/GPU cores and allowing for a larger batch size.

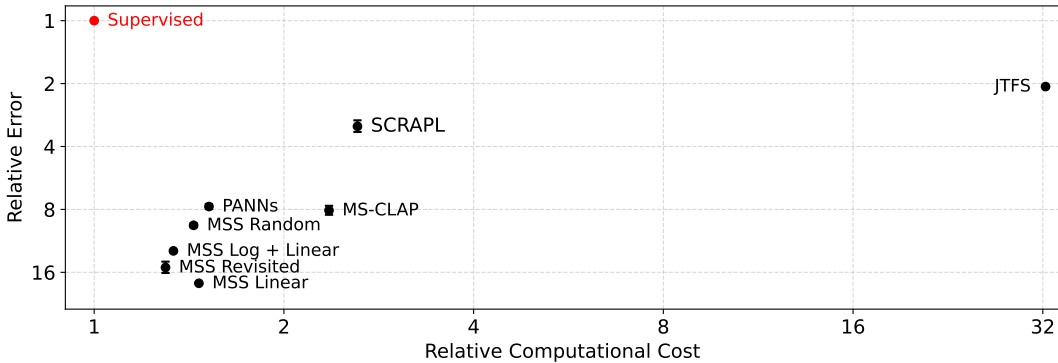

Figure 1: Mean average synthesizer parameter error (y-axis) versus computational cost (x-axis) of unsupervised sound matching models for the granular synthesis task. Both axes are rescaled by the performance of a supervised model with the same number of parameters. Whiskers denote 95% CI, estimated over 20 random seeds. Due to computational limitations, JTFS-based sound matching is evaluated only once.

## 2 RELATED WORK

The guiding intuition behind SCRAPL is that natural signals and images exhibit strong correlations across ST paths. This fact has been observed empirically since the onset of ST research (Bruna & Mallat, 2013; Andén & Mallat, 2011) and aligns with earlier work on texture modeling based on pairwise correlations between wavelet modulus coefficients (Portilla & Simoncelli, 2000).

Visual and auditory textures, understood as stationary random fields, play a key role in applied ST research. ST features outperform short-term Fourier features (e.g., MSS) in their ability to characterize intermittency in non-Gaussian textures (Muzy et al., 2015). Texture resynthesis by gradient descent of ST loss has been applied to such diverse settings as computer music creation (Lostanlen et al., 2019) and the study of the cosmic microwave background (Delouis et al., 2022).

The democratization of differentiable programming toolkits (e.g., TensorFlow, PyTorch, JAX) has greatly advanced the flexibility of gradient backpropagation in "hybrid" scattering–neural networks involving learnable and non-learnable modules. Angles & Mallat (2018) have built a hybrid scattering–GAN model for image generation, in which ST distance plays the role of a discriminator.

To our knowledge, the closest prior work to SCRAPL is the pruned graph scattering transform (pGST) of Ioannidis et al. (2020), a method which reduces the complexity of ST by pruning the path set $\mathscr{P}$ down to a proper subset $\mathscr{P}' \subset \mathscr{P}$, based on a graph-spectrum-inspired criterion. Although both pGST and SCRAPL share a similar overarching goal, let us point out that pGST is a feature selection method: the cardinality of $\mathscr{P}'$ is typically $\sim 10\%$ that of $\mathscr{P}$ and $\mathscr{P}'$ is kept fixed across training examples and across epochs. In comparison, SCRAPL performs a more radical pruning, down to a single path (card $\mathscr{P}' = 1$), while harnessing dedicated techniques in stochastic optimization ($\mathscr{P}$-Adam and $\mathscr{P}$-SAGA) to reduce the variance of ST loss during gradient backpropagation.

## 3 METHODS

### 3.1 STOCHASTIC APPROXIMATION OF SCATTERING TRANSFORM LOSS GRADIENT

The proposition below, proven in Appendix A, shows that if paths are drawn uniformly at random, the stochastic approximation in SCRAPL is unbiased: in other words, the expected value of the stochastic gradient of per-path loss is equal to the gradient of full ST loss.

**Proposition 3.1.** Let $\Phi = (\phi_p)_0^{P-1}$ be a scattering transform with $P$ paths. Given a signal or image $\boldsymbol{x}$, let $F_{\boldsymbol{x}}$ be an autoencoder operating on $\boldsymbol{x}$ and let $\mathscr{L}_{\boldsymbol{x}}^{\Phi}$ be the associated ST reconstruction loss. Let $\mathscr{U}_P$ be the uniform distribution over $\mathscr{P} = \{0, \ldots, P-1\}$. One has, for every weight vector $\boldsymbol{w}$:

$$\mathbb{E}_{z \sim \mathscr{U}_P}\left[\boldsymbol{\nabla}(\mathscr{L}_{\boldsymbol{x}}^{\phi_z} \circ F_{\boldsymbol{x}})(\boldsymbol{w})\right] = \boldsymbol{\nabla}(\mathscr{L}_{\boldsymbol{x}}^{\Phi} \circ F_{\boldsymbol{x}})(\boldsymbol{w}). \tag{5}$$

Although a uniform sampling of paths matches the intuition of approximating the ST gradient in expectation, we will see that this may be suboptimal. The $\theta$-importance sampling method, which we will present in Section 3.4, does not satisfy the hypothesis of Proposition 3.1; yet, it consistently outperforms uniform sampling as part of SCRAPL. The design of biased stochastic approximation schemes is an active topic in machine learning research (Dieuleveut et al., 2023).

## 3.2 $\mathscr{P}$-ADAM: PATH-WISE ADAPTIVE MOMENT ESTIMATION

The key idea behind the Adam optimizer is to smooth the successive realizations of the stochastic gradient, here denoted by $g$, via autoregressive estimates of its first- and second-order element-wise moments, denoted by $m$ and $v$ (Kingma & Ba, 2014). However, the smoothing technique in Adam is ineffective for SCRAPL because the gradients of path-wise ST losses are not identically distributed. Against this problem, we propose to maintain $P$ estimates of path-wise moments ($\mathscr{P}$-Adam):

$$m_p \leftarrow \beta_1^{(k-\tau_p)/P} m_p + (1 - \beta_1^{(k-\tau_p)/P}) g \tag{6}$$

$$v_p \leftarrow \beta_2^{(k-\tau_p)/P} v_p + (1 - \beta_2^{(k-\tau_p)/P})(g \odot g), \tag{7}$$

where $k$ is the current iteration number, $\tau_p$ is the iteration when path $p$ was last drawn; $\beta_1$ and $\beta_2$ are hyperparameters; and the circled dot denotes element-wise multiplication of vectors. The exponent $(k - \tau_p)/P$ adapts the time constant of smoothing to the recency of the previous estimate.

The second step in $\mathscr{P}$-Adam, following classical Adam, consists of bias correction and the ratio of debiased first-order moment to stable square root of debiased second-order moment:

$$g_{\text{current}} = \frac{\dfrac{m_p}{1 - \beta_1^{k/P}}}{\sqrt{\varepsilon + \dfrac{v_p}{1 - \beta_2^{k/P}}}}, \tag{8}$$

where we have adapted the original exponents of Adam ($\beta_1^k$, $\beta_2^k$) to account for the number of paths.

## 3.3 $\mathscr{P}$-SAGA: PATH-WISE STOCHASTIC AVERAGE GRADIENT WITH ACCELERATION

The stochastic average gradient (SAG) algorithm has the potential to accelerate stochastic gradient descent in the context of the minimization of finite sums (Schmidt et al., 2017). Although this sum is typically over training examples in neural network training, in SCRAPL, Equation 3 is a sum over paths for a given example $x$. With this observation in mind, we propose $\mathscr{P}$-SAGA, a path-wise version of SAG with acceleration (SAGA, Defazio et al. (2014)). We maintain a memory of the last $\mathscr{P}$-Adam updates over each path, denoted by $(\hat{g}_p)_0^{P-1}$; and the set of paths previously visited, denoted by $\Gamma$. Given a learning rate $\alpha_k$ at iteration $k$, the $\mathscr{P}$-SAGA update is:

$$w \leftarrow w - \alpha_k \left( g_{\text{current}} - \hat{g}_p + \frac{\sum_{\gamma \in \Gamma} \hat{g}_\gamma}{\max(1, \text{card}\,\Gamma)} \right). \tag{9}$$

Unlike the original SAG and SAGA algorithms, $\mathscr{P}$-SAGA's additional memory footprint is proportional to $P$, not the size of the dataset $N$, making it suitable for neural network training and real-world optimization tasks like our experiments in Section 4. We also note that $\mathscr{P}$-Adam and $\mathscr{P}$-SAGA introduce no additional hyperparameters over the standard Adam optimizer. Algorithm 1 summarizes SCRAPL with both $\mathscr{P}$-Adam and $\mathscr{P}$-SAGA enabled.

## 3.4 $\theta$-IMPORTANCE SAMPLING

We now consider the important special case of differentiable digital signal processing (DDSP, see Section 1), in which the autoencoder composes a non-learnable decoder, typically a synthesizer, with a learned encoder: i.e., $F_x = (D \circ E_x)$ with $E_x(w) = \tilde{\theta}$ and $D(\tilde{\theta}) = \tilde{x}$ (Engel et al., 2020). We assume both $D$ and $E_x$ to be differentiable with respect to their inputs, but $D$ is not necessarily deterministic. We denote by $U$ the dimension of the parameter space $\theta$; i.e., the output space of $E_x$ and input space of $D$.

---

**Algorithm 1** "**Sc**attering transform with **Ra**ndom **P**aths for machine **L**earning" (SCRAPL). The pseudo-code below describes SCRAPL with a batch size equal to one, without loss of generality.

---

**Require:** $\Phi = (\phi_p)_0^{P-1}$: Scattering transform (ST) with $P$ paths
**Require:** $\pi$: Categorical distribution over the path set $\mathscr{P} = \{0, \dots, P-1\}$
**Require:** $F$: Autoencoder with trainable parameters $\boldsymbol{w}$
**Require:** $\boldsymbol{w}$: Neural network weights at initialization
**Require:** $\beta_1, \beta_2, \varepsilon$: Adam hyperparameters
**Require:** $(\alpha_k)_1^K$: Learning rate schedule
  $\Gamma \leftarrow \emptyset$
  **for** $p$ in $\{0, \dots, P-1\}$ **do**
    $\tau_p \leftarrow 0$
    $\boldsymbol{m}_p \leftarrow \boldsymbol{0}$
    $\boldsymbol{v}_p \leftarrow \boldsymbol{0}$
    $\hat{\boldsymbol{g}}_p \leftarrow \boldsymbol{0}$
  **end for**
  **for** $k$ in $\{1, \dots, K\}$ **do**

> $n \leftarrow$ draw an integer uniformly at random in $\{0, \dots, N-1\}$
> $p \leftarrow$ draw an integer at random in $\{0, \dots, P-1\}$ according to $\pi$    {Stochastic approx.}
> $\mathscr{L}(\boldsymbol{w}) \leftarrow P\|\phi_p(\boldsymbol{x}_n) - (\phi_p \circ F_{\boldsymbol{w}})(\boldsymbol{x}_n)\|_2^2$
> $\boldsymbol{g} \leftarrow \nabla\mathscr{L}(\boldsymbol{w})$

> $\boldsymbol{m}_p \leftarrow \beta_1^{(k-\tau_p)/P}\boldsymbol{m}_p + (1 - \beta_1^{(k-\tau_p)/P})\boldsymbol{g}$
> $\boldsymbol{v}_p \leftarrow \beta_2^{(k-\tau_p)/P}\boldsymbol{v}_p + (1 - \beta_2^{(k-\tau_p)/P})(\boldsymbol{g} \odot \boldsymbol{g})$
> $\hat{\boldsymbol{m}} \leftarrow \boldsymbol{m}_p/(1 - \beta_1^{k/P})$
> $\hat{\boldsymbol{v}} \leftarrow \boldsymbol{v}_p/(1 - \beta_2^{k/P})$    {$\mathscr{P}$-Adam}
> $\boldsymbol{g}_{\text{current}} \leftarrow \hat{\boldsymbol{m}}/\sqrt{\varepsilon + \hat{\boldsymbol{v}}}$
> $\tau_p \leftarrow k$

> $\boldsymbol{g}_{\text{avg}} \leftarrow \dfrac{1}{\max(1, \operatorname{card}\Gamma)}\sum_{\gamma \in \Gamma}\hat{\boldsymbol{g}}_\gamma$
> $\boldsymbol{g}_{\text{SAGA}} \leftarrow \boldsymbol{g}_{\text{current}} - \hat{\boldsymbol{g}}_p + \boldsymbol{g}_{\text{avg}}$    {$\mathscr{P}$-SAGA}
> $\boldsymbol{w} \leftarrow \boldsymbol{w} - \alpha_k\boldsymbol{g}_{\text{SAGA}}$
> $\hat{\boldsymbol{g}}_p \leftarrow \boldsymbol{g}_{\text{current}}$
> $\Gamma \leftarrow \Gamma \cup \{p\}$

  **end for**
  **return** $\boldsymbol{w}$

---

A known drawback of DDSP is that the optimization landscape of spectral loss in parameter space (i.e., of $\mathscr{L}_{\boldsymbol{x}}^{\Phi} \circ D$) may not coincide with that of supervised parameter loss (i.e., Euclidean distance to $\boldsymbol{\theta}$, also known as P-loss) (Hayes et al., 2024). Against this drawback, we propose a method named $\theta$-importance sampling ($\theta$-IS), which constructs a categorical distribution $\pi$ over the path space $\mathscr{P}$. The key idea behind $\theta$-IS is to introduce bias in the stochastic approximation of spectral loss so as to bring it closer to P-loss. For lack of supervision, we are unable to construct the optimal distribution $\pi$ but provide a heuristic of this form:

$$\pi_p = \frac{1}{U}\sum_{u=0}^{U-1}\frac{C_{u,p}}{\sum_{p=0}^{P-1}C_{u,p'}}, \tag{10}$$

where, intuitively, $C_{u,p}$ represents the importance of parameter dimension $\theta_u$ upon path $p$. We rescale this importance relative to all paths and average uniformly across parameters $u$, yielding an importance-weighted categorical distribution $\pi$ over paths. We then use $\pi$ instead of a uniform distribution for sampling paths in the SCRAPL algorithm (see Algorithm 1).

Let $E_{\boldsymbol{x},u}(\boldsymbol{w})$ denote the $u^{\text{th}}$ coordinate of $E_{\boldsymbol{x}}(\boldsymbol{w})$. Given $\boldsymbol{w}$, we measure the sensitivity of each ST path $p$ to the parameter control $u$ around the input $\boldsymbol{x}$ in terms of the following partial derivative:

$$s_{\boldsymbol{x},u,p} : \boldsymbol{w} \longmapsto \frac{\partial\left(\mathscr{L}_{\boldsymbol{x}}^{\phi_p} \circ D\right)}{\partial\theta_u}\left(E_{\boldsymbol{x},u}(\boldsymbol{w})\right) \tag{11}$$

To convert the sensitivity function $s_{\boldsymbol{x},u,p}$ into relative importance $C_{u,p}$, we analyze the curvature of the loss landscape $\mathscr{L}_{\boldsymbol{x}}^{\phi_p}$. We multiply $s_{\boldsymbol{x},u,p}$ by the gradient of $E_{\boldsymbol{x},u}$, yielding a linear transformation of neural network parameters. The coordinate-wise gradient of this linear transformation yields a positive definite matrix: we compute its largest eigenvalue. We repeat this process over a representative dataset $\mathscr{X}$ of $N_{\text{IS}}$ unlabeled signals from the training dataset. Formally:

$$C_{u,p} = \mathbb{E}_{\boldsymbol{x} \sim \mathscr{X}} \left[ \lambda_{\max} \left( \boldsymbol{\nabla}_{\boldsymbol{w}} \left( s_{\boldsymbol{x},u,p}(\boldsymbol{w}) \boldsymbol{\nabla} E_{\boldsymbol{x},u}(\boldsymbol{w}) \right) \right) \right], \tag{12}$$

where $\lambda_{\max}(\mathbf{M})$ is the magnitude of the largest eigenvalue of a square matrix $\mathbf{M}$; $\boldsymbol{\nabla}_{\boldsymbol{w}}$ is the gradient with respect to $\boldsymbol{w}$. In practice, we compute $\lambda_{\max}(\mathbf{M})$ using a stochastic power iteration with deflation and the Hessian vector product, which has the same asymptotic runtime complexity as a backpropagation step.[1] Crucially, the computation required for $\theta$-IS can be trivially parallelized across $p$ and $u$, and only needs to be computed once before training.

Our definition of $\theta$-IS is inspired by Schmidt et al. (2017), who propose a variant of the SAG algorithm in which mini-batches are sampled non-uniformly; more precisely, in proportion to the Lipschitz constant of the gradients, which we approximate using Equation 12 for each $p$ and $u$. This heuristic relies on the argument that gradients which change quickly should be regarded as more important than gradients which change slowly.

## 4 EXPERIMENTS

We apply SCRAPL to a differentiable implementation of the joint time–frequency scattering transform (Muradeli et al., 2022). We conduct three unsupervised sound matching experiments under the DDSP autoencoder paradigm described in Section 3.4. The encoder, $E_{\boldsymbol{x}}$, is a convolutional neural network which operates on a constant-$Q$ transform (Cheuk et al., 2020). We use relatively lightweight neural networks for our experiments, a choice made possible by the strong inductive bias DDSP provides and informed by prior work (Han et al., 2025) indicating that larger networks do not necessarily improve sound matching capabilities. We choose all hyperparameters in experiments heuristically.

To highlight the new kinds of perceptual quality assessment tasks SCRAPL enables, all three experiments investigate nondeterministic decoders that introduce random time shifts into the resulting reconstructed audio. While our experiments are for a discriminative and generative audio processing task, we emphasize that SCRAPL is a general algorithm and can be equally applied to deep inverse problems that leverage other scattering transforms.

### 4.1 JOINT TIME–FREQUENCY SCATTERING TRANSFORM (JTFS)

The joint time–frequency scattering transform (JTFS) is a nonlinear convolutional operator which extracts spectrotemporal modulations over the constant-$Q$ spectrogram (Andén et al., 2019). The multivariable filter $\boldsymbol{\Psi}_p$ comprises two stages: temporal scattering, i.e., 1-D band-pass filtering with Morlet wavelets over the time axis; and frequential scattering, i.e., idem over the log-frequency axis. The center frequencies of band-pass filters for temporal scattering, called *rates*, are measured in Hertz. The center frequencies for frequential scattering, called *scales*, are measured in cycles per octave. Thus, in the case of JTFS, the path index $p$ is a rate–scale multiindex.

The JTFS has been shown to correlate with human perception (Lostanlen et al., 2021; Tian et al., 2025) and can provide an informative gradient for audio comparisons that are misaligned (Vahidi et al., 2023) or benefit from multi-resolution analysis like percussive sounds (Han et al., 2024), which is why we select it as the underlying ST of the SCRAPL algorithm in our unsupervised sound matching experiments. Additionally, due to its computational complexity, until now it has been used almost exclusively as a precomputed feature instead of a loss function for neural network training.

### 4.2 GRANULAR SYNTH SOUND MATCHING

Granular synthesis is an example of a new class of synths that can be effectively sound matched with SCRAPL and the JTFS, due to its inherently stochastic audio generation process with individual grains being misaligned in time at the micro-level, but still being perceived as a single texture. It has been extensively used in the production of electronic music since the late 1950s[2] and played a

---

[1] https://github.com/noahgolmant/pytorch-hessian-eigenthings/
[2] https://www.iannis-xenakis.org/en/granular-synthesis/

fundamental role in the creation of contemporary electronic music genres. Our differentiable granular synth produces textures of chirplet grains with random temporal positions, center frequencies, and chirp rates, and has two continuous parameters: density ($\theta_{\text{density}}$) which controls how many grains are produced, and slope ($\theta_{\text{slope}}$) which controls their rate of frequency modulation.

We compare four MSS-based losses: linear, log + linear (Engel et al., 2020), random (Steinmetz & Reiss, 2020), and a SOTA hyperparameter-tuned revisited MSS loss (Schwär & Müller, 2023). Given their correlation with human perception (Kilgour et al., 2019; Tailleur et al., 2024), we also include the Euclidean distance of MS-CLAP (Elizalde et al., 2023) and PANNs Wavegram Logmel embeddings (Kong et al., 2020). In addition, we train with ordinary (i.e., full-tree) JTFS so as to put the speed and accuracy of SCRAPL into context. Lastly, as an estimate of best achievable performance with our encoder architecture and training configuration, we run a supervised version of sound matching, also known as "parameter loss" or P-loss for short (see Section 3.4). We summarize the implementation details in Appendix E.

## 4.3   Chirplet Synth Sound Matching

Similar to the unsupervised granular synth sound matching experiment, we evaluate our $\theta$-importance sampling initialization heuristic for SCRAPL on a differentiable chirplet synth (based on the implementation by Vahidi et al. (2023)) with two parameters: $\theta_{\text{AM}}$ which controls the rate of amplitude modulation (Hz) and $\theta_{\text{FM}}$ which controls the rate of frequency modulation (oct/s). Since the paths in the JTFS correspond to specific wavelet AM and FM center frequencies, given a chirplet synth configuration with bounded $\theta_{\text{AM}}$ and $\theta_{\text{FM}}$ ranges, we know approximately which paths of the JTFS should provide the most informative gradients for the synth parameters. After computing our initialization heuristic, we can analyze the resulting path probabilities and verify that the paths within the parameter ranges of the synth have been assigned a probability greater than uniform.

We evaluate four different synth configurations:

1. Slow AM ($\theta_{\text{AM}} \in [1.0, 2.0]$ Hz), slow FM ($\theta_{\text{FM}} \in [0.5, 1.0]$ oct/s);
2. Slow AM ($\theta_{\text{AM}} \in [1.0, 2.0]$ Hz), moderate FM ($\theta_{\text{FM}} \in [2.0, 4.0]$ oct/s);
3. Fast AM ($\theta_{\text{AM}} \in [2.8, 8.4]$ Hz), moderate FM ($\theta_{\text{FM}} \in [2.0, 4.0]$ oct/s);
4. Fast AM ($\theta_{\text{AM}} \in [2.8, 8.4]$ Hz), fast FM ($\theta_{\text{FM}} \in [4.0, 12.0]$ oct/s).

We compare SCRAPL training runs using uniform sampling and $\theta$-importance sampling calculated from a single training batch of 32 examples. We summarize the implementation details in Appendix E.

## 4.4   Roland TR-808 Sound Matching

As a real-world evaluation task, we sound match a DDSP implementation (Shier et al., 2024) of the Roland TR-808 Rhythm Composer drum machine, a historically meaningful synthesizer for the creation of Detroit techno, house, and hip-hop music.[3] Inharmonic transient sounds like percussion are a form of non-stationary signal that the JTFS is well suited for perceptual quality assessment (Han et al., 2024) because of its ability to extract spectrotemporal patterns at multiple scales and rates. Additionally, due to the transient nature of drum sounds, they are highly sensitive to even a few milliseconds of misalignment, thus further benefiting from the time invariance of JTFS.

We use a high fidelity, 100% analog dataset[4] of 681 bass drum, snare, tom, and hi-hat one-shot recordings of the TR-808 and repeat experiments 40 times on different train/validation/test splits and random seeds. Since the transient of analog drum recordings is rarely perfectly aligned, and no two analog TR-808 drum synths produce the same signal, we investigate both perfectly aligned (labeled *micro*) and unaligned (labeled *meso*) sound matching by up to $\pm46$ ms ($\pm2048$ samples at 44.1 kHz). Given its correlation with human perception, we employ the JTFS and Fréchet Audio Distance (FAD) (Kilgour et al., 2019; Défossez et al., 2023) as evaluation metrics. We also include MSS and mean frame-by-frame perceptual loudness and loudness-weighted perceptually-scaled spectral centroid and flatness for both the transient and decay portions of reconstructed signals (nine metrics in total). Additional context for these last six metrics can be found in Shier et al. (2024). We summarize the implementation details in Appendix E.

---

[3] https://www.roland.com/global/promos/roland_tr-808/
[4] https://samplesfrommars.com/products/tr-808-samples/

Table 1: Evaluation results for the unsupervised granular synth sound matching task with two continuous $\theta_{\text{synth}}$ parameters: $\theta_{\text{density}}$ and $\theta_{\text{slope}}$ (more details in Section 4.2). Uncertainties are 95% CI for 20 training runs using different random seeds. Due to computational limitations, the JTFS method is only evaluated once.

| Method | $\theta_{\text{synth}}\ L_1$ ‰↓ | | $\theta_{\text{density}}\ L_1$ ‰↓ | | $\theta_{\text{slope}}\ L_1$ ‰↓ | |
|---|---|---|---|---|---|---|
| JTFS | **42.4** | | **65.8** | | **19.0** | |
| SCRAPL (no $\theta$-IS) | 73.8 | $\pm 13$ | 70.4 | $\pm\ 8.8$ | 77.2 | $\pm 19$ |
| SCRAPL | 65.7 | $\pm\ 4.2$ | 72.6 | $\pm\ 6.3$ | 58.7 | $\pm\ 7.5$ |
| MSS Linear | 370 | $\pm\ 0.52$ | 499 | $\pm\ 0.84$ | 241 | $\pm\ 0.28$ |
| MSS Log + Linear | 259 | $\pm\ 1.7$ | 277 | $\pm\ 3.2$ | 241 | $\pm\ 0.42$ |
| MSS Revisited | 311 | $\pm 19$ | 376 | $\pm 40$ | 246 | $\pm\ 3.0$ |
| MSS Random | 195 | $\pm\ 4.2$ | 149 | $\pm\ 7.8$ | 242 | $\pm\ 1.0$ |
| MS-CLAP | 166 | $\pm\ 8.2$ | 81.9 | $\pm\ 9.0$ | 250 | $\pm\ 8.2$ |
| PANNs Wavegram-Logmel | 159 | $\pm\ 4.4$ | 80.3 | $\pm\ 4.2$ | 238 | $\pm\ 5.5$ |
| P-loss | 20.5 | $\pm\ 0.20$ | 24.7 | $\pm\ 0.31$ | 16.3 | $\pm\ 0.31$ |

## 5 RESULTS

### 5.1 GRANULAR SYNTH SOUND MATCHING

We benchmark all loss function computational costs (see Appendix B, Table 5) and plot them in Figure 1 against their evaluation accuracy (see Table 1) on $\theta_{\text{synth}}\ L_1$ relative to supervised training (i.e., P-loss). We observe that SCRAPL comes within factor two of JTFS in terms of accuracy, and within factor two of MSS in terms of runtime, striking a notable balance between them. The significant difference in runtime and convergence between JTFS and SCRAPL is further illustrated in Figure 2 where we plot validation accuracy against wall-clock time, instead of optimization steps. We also note that MSS is unable to sound match the synth at all, and the SOTA embedding losses are only able to optimize $\theta_{\text{density}}$, albeit not as well as SCRAPL and JTFS. Validation accuracy curves for all methods are also provided in Figure 2. We provide a variation of Figure 1 plotting the test JTFS audio distance on the y-axis in Appendix B, Figure 3.

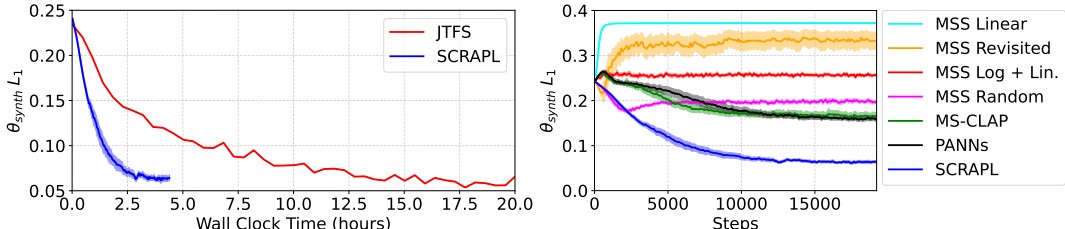

Figure 2: Left: JTFS vs. SCRAPL wall-clock training times on a single NVIDIA RTX A5000 GPU. Due to computational limitations, the JTFS method is only evaluated once. Right: Validation convergence graphs for the unsupervised granular synth sound matching task. Both: Shaded areas are 95% CI for 20 training runs using different random seeds.

Table 2 summarizes the results of an ablation of SCRAPL and its $\mathscr{P}$-Adam, $\mathscr{P}$-SAGA, and $\theta$-IS optimization techniques for the granular synth sound matching task, with Appendix B, Table 6 providing additional information about statistical significance. There is a clear monotonic improvement in accuracy and convergence time for each technique, as well as a reduction in variance provided by $\mathscr{P}$-SAGA, and $\theta$-IS. We emphasize that SCRAPL without any extra optimization techniques still outperforms all other non-JTFS methods in terms of accuracy, demonstrating its ability to optimize a new class of problems and making just stochastic sampling of scattering transforms a viable approach if the additional memory and computational requirements of $\mathscr{P}$-Adam, $\mathscr{P}$-SAGA, and $\theta$-IS are undesirable. Finally, from Table 1, we see that $\theta$-IS results in a better overall accuracy of $\theta_{\text{synth}}$ than uniform sampling (despite $\theta_{\text{density}}$ now being slightly worse), which is consistent with our hypothesis from Section 3.4 that $\theta$-IS results in more balanced convergence of all synth parameters. Validation accuracy curves for all ablations are provided in Appendix B, Figure 4.

Table 2: Ablation table for SCRAPL with test results and validation $\theta_{\text{synth}} L_1$ total variation and convergence steps for the unsupervised granular synth sound matching task. Convergence is defined as $\theta_{\text{synth}} L_1 < 100\,\text{‰}$. Statistical significance results for each additional optimization technique are presented in Appendix B, Table 6. Uncertainties are 95% CI for 20 training runs using different random seeds. Due to computational limitations, the JTFS method is only evaluated once.

| Method | $\mathscr{P}$-Adam | $\mathscr{P}$-SAGA | $\theta$-IS | Test $\theta_{\text{synth}} L_1$ ‰ ↓ | Validation Total Var. ↓ | Conv. Steps ↓ |
|---|---|---|---|---|---|---|
| SCRAPL | | | | 99.7 ± 8.2 | 5.30± 0.25 | 10 906±1170 |
| | ✓ | | | 87.4 ±15 | 6.98± 0.25 | 8006± 697 |
| | ✓ | ✓ | | 73.8 ±13 | 3.46± 0.15 | 7296± 683 |
| | ✓ | ✓ | ✓ | **65.7** ± 4.2 | **3.27**± 0.12 | **6014**± 642 |
| JTFS | | | | 42.4 | 5.66 | 1442 |
| P-loss | | | | 20.5 ± 0.20 | 1.83± 0.025 | 672± 23 |

In summary, this experiment demonstrates that the variance of the gradient elicited by the stochastic approximation of a ST with the SCRAPL algorithm is manageably small in the context of deep neural network (DNN) training, resulting in a favorable tradeoff between computational speed and convergence rate when compared to full-tree scattering (i.e., the JTFS). Training with SCRAPL is nearly equivalent to training with the gradient of full-tree scattering in terms of JTFS loss on unseen test data: see Appendix B, Figure 3. In terms of synthesizer parameter error, training with SCRAPL is not as accurate as training with full-tree ST, but outperforms prior work: see Table 1. This SOTA result paves the way towards a new kind of DNN training for deep inverse problems, in which the forward operator (synth) produces nondeterministic time–frequency patterns.

Understanding the convergence properties of algorithms like SCRAPL for convex and non-convex tasks is an active area of research (Reddi et al., 2016; 2018; Défossez et al., 2022; Kim & Oh, 2025). Our proof of Proposition 3.1 in Appendix A that SCRAPL without any additional optimization techniques is an unbiased estimator of full-tree ST is an important first step in this direction. We believe that further convergence analysis of SCRAPL remains a promising avenue for future work.

## 5.2 CHIRPLET SYNTH SOUND MATCHING

Table 3 summarizes the chirplet synth evaluation results, with Appendix C, Figure 5 showing validation accuracy curves for uniform and $\theta$-importance sampling on the four synth configurations. $\theta$-IS improves the prediction of $\theta_{\text{AM}}$ by 25–55% and of $\theta_{\text{FM}}$ by 14–80%, while reducing time to convergence by 23–50%: see Appendix C, Table 7. Of course, these improvements are for synth configurations that have been designed to showcase the benefit of nonuniform sampling of paths; however, this overall trend remains true, albeit not as pronounced, for the granular synth (Table 2) and real-world sound matching task (Table 4). Finally, we plot the path $\theta$-IS probabilities in Appendix C, Figure 6 and observe that indeed, a unique distribution is learned for each synth, and the greater than uniform probabilities appear to roughly correspond to each configuration's limited AM / FM range.

## 5.3 ROLAND TR-808 SOUND MATCHING

Table 4 and Appendix D, Tables 8, and 9 summarize the unsupervised Roland TR-808 synth sound matching audio distance, transient, and decay perceptual similarity results. Overall, we observe that JTFS dominates almost all metrics in both micro and meso environments, showcasing its suitability for transient percussive sounds and temporal invariance. After JTFS, MSS performs best when samples are perfectly aligned (micro), but performs worse in the unaligned (meso) setting and is unable to match the transient, which is the most salient part of a drum hit. In contrast, SCRAPL shows consistent sound matching performance in both micro and meso environments and is able to preserve the transient even when audio is misaligned. However, SCRAPL fails to recover the less audible decay portion of the signal. We hypothesize this is due to informative, low-frequency paths for the decay being sparse and underrepresented in the categorical distribution over paths, even after accounting for $\theta$-IS. We provide listening samples at the accompanying website[5] and encourage readers to evaluate the results directly.

---

[5]Companion website: `https://christhetree.github.io/scrapl/`

Table 3: Evaluation results for SCRAPL with and without the $\theta$-importance sampling initialization heuristic on unsupervised sound matching of four different AM / FM chirplet synths with two continuous $\boldsymbol{\theta}_{\text{synth}}$ parameters: $\theta_{\text{AM}}$ and $\theta_{\text{FM}}$ (more details in Section 4.3). Uncertainties are 95% CI for 20 training runs using different random seeds.

| Sampling Method | Synth Configuration | | $\theta_{\text{AM}}\ L_1\ \text{‰}\downarrow$ | | $\theta_{\text{FM}}\ L_1\ \text{‰}\downarrow$ | |
|---|---|---|---|---|---|---|
| $(\pi)$ | $\theta_{\text{AM}}$ (Hz) | $\theta_{\text{FM}}$ (oct/s) | | | | |
| Uniform | $1.0 - 2.0$ | $0.5 - 1.0$ | 124 | $\pm 10$ | 155 | $\pm 18$ |
| $\theta$-IS | | | **77.7** | $\pm\ 6.7$ | **78.4** | $\pm 11$ |
| Uniform | $1.0 - 2.0$ | $2.0 - 4.0$ | 111 | $\pm 20$ | 68.6 | $\pm 11$ |
| $\theta$-IS | | | **55.5** | $\pm\ 4.1$ | **44.4** | $\pm\ 2.8$ |
| Uniform | $2.8 - 8.4$ | $2.0 - 4.0$ | 122 | $\pm 22$ | 238 | $\pm 21$ |
| $\theta$-IS | | | **54.9** | $\pm\ 3.5$ | **48.5** | $\pm\ 4.7$ |
| Uniform | $2.8 - 8.4$ | $4.0 - 12.0$ | 108 | $\pm 12$ | 95.6 | $\pm 20$ |
| $\theta$-IS | | | **81.5** | $\pm 12$ | **82.1** | $\pm 11$ |

Table 4: Audio distance evaluation results for the unsupervised Roland TR-808 DDSP synth sound matching task with 14 continuous $\boldsymbol{\theta}_{\text{synth}}$ parameters (more details in Section 4.4). Uncertainties are 95% CI for 40 training runs using different random seeds and dataset splits. Due to computational limitations, the JTFS method is only trained and evaluated for 4 random seeds.

| Method | MSS Log. + Linear $\downarrow$ | | JTFS $\downarrow$ | | FAD (EnCodec) $\downarrow$ | |
|---|---|---|---|---|---|---|
| | Micro | Meso | Micro | Meso | Micro | Meso |
| JTFS | $617 \pm 46$ | $622 \pm 45$ | **490** $\pm 28$ | **523** $\pm 17$ | **0.781** $\pm 0.069$ | **1.04** $\pm 0.15$ |
| SCRAPL | | | | | | |
| (no $\theta$-IS) | $862 \pm 36$ | $944 \pm 48$ | $1140 \pm 48$ | $1250 \pm 51$ | $2.75\ \pm 0.39$ | $2.85 \pm 0.38$ |
| SCRAPL | $857 \pm 42$ | $879 \pm 42$ | $1050 \pm 50$ | $1110 \pm 52$ | $2.43\ \pm 0.22$ | $2.42 \pm 0.22$ |
| MSS Lin. | $611 \pm 15$ | $724 \pm 37$ | $779 \pm 31$ | $1470 \pm 83$ | $1.22\ \pm 0.082$ | $3.33 \pm 0.46$ |
| MSS L+L | **596** $\pm 19$ | **615** $\pm 18$ | $1260 \pm 58$ | $1390 \pm 49$ | $2.14\ \pm 0.39$ | $3.01 \pm 0.40$ |
| MSS Rev. | $637 \pm 16$ | $797 \pm 20$ | $870 \pm 23$ | $1250 \pm 27$ | $2.02\ \pm 0.37$ | $2.21 \pm 0.34$ |
| MSS Rand. | $682 \pm 25$ | $700 \pm 26$ | $1410 \pm 87$ | $1500 \pm 59$ | $7.03\ \pm 2.2$ | $6.65 \pm 1.7$ |

## 6 CONCLUSION

Differentiable similarity measures have the potential to enhance the perceptual quality of generative models and deep inverse problem solvers. In spite of their mathematical guarantees and neurophysiological plausibility, scattering transforms (ST) have not been able to realize this potential, for lack of tractable optimization algorithms. To fill this gap, SCRAPL takes advantage of the tree-like structure of ST to save computation at each backward pass. Our numerical simulations show the value of SCRAPL for unsupervised sound matching, particularly when the synthesizer of interest is nondeterministic. Although our ST architecture of choice is joint time–frequency scattering (JTFS), we stress that SCRAPL is agnostic to the specifics of multivariable filterbank design: beyond wavelet scattering, it extends to learnable scattering-like architectures (Lattner et al., 2019; Cotter & Kingsbury, 2019; Gauthier et al., 2022). We consider investigating SCRAPL's generalizability to other ST architectures, different audio tasks such as speech enhancement and automatic mixing, and additional modalities like adversarial image generation and texture synthesis as promising directions for future work. As a longer-term perspective, the success of our architecture-informed importance sampling heuristic highlights the opportunity to meta-learn the relative importance of each ST path for the task at hand over the course of neural network training (Yamaguchi et al., 2023).

REPRODUCIBILITY STATEMENT

Appendix E contains hyperparameters and training details for each of the three experiments in this paper. We also provide listening samples, source code, configuration files, instructions to reproduce our experiments, and the SCRAPL algorithm as a Python package at the companion website: `https://christhetree.github.io/scrapl/`

ACKNOWLEDGMENTS

Christopher Mitcheltree is a research student at the UKRI Centre for Doctoral Training in Artificial Intelligence and Music, supported jointly by UK Research and Innovation (grant number EP/S022694/1) and Queen Mary University of London. This research was funded in part by the French National Research Agency (ANR) under the project MuReNN "ANR-23-CE23-0007".

C.M. would like to thank Jordie Shier for his help with the differentiable Roland TR-808 synthesizer implementation and the insightful conversations about applying and evaluating the SCRAPL algorithm.

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

## A  PROOF OF PROPOSITION 3.1

Let us write $\tilde{\boldsymbol{x}} = F_{\boldsymbol{x}}(\boldsymbol{w})$. By linearity of the gradient, we may decompose $\boldsymbol{\nabla}\mathscr{L}_{\boldsymbol{x}}^{\Phi}(\tilde{\boldsymbol{x}})$ over paths:

$$\boldsymbol{\nabla}\mathscr{L}_{\boldsymbol{x}}^{\Phi}(\tilde{\boldsymbol{x}}) = \frac{1}{P}\sum_{p=0}^{P-1}\boldsymbol{\nabla}\mathscr{L}_{\boldsymbol{x}}^{\phi_p}(\tilde{\boldsymbol{x}}). \tag{13}$$

Let us denote the Jacobian of $F_{\boldsymbol{x}}$ at $\boldsymbol{w}$ by $\mathbf{J}_{F_{\boldsymbol{x}}}(\boldsymbol{w})$. For each $p \in \mathscr{P}$, we apply the chain rule:

$$\boldsymbol{\nabla}(\mathscr{L}_{\boldsymbol{x}}^{\phi_p} \circ F_{\boldsymbol{x}})(\boldsymbol{w}) = \mathbf{J}_{F_{\boldsymbol{x}}}(\boldsymbol{w})^{\top}\boldsymbol{\nabla}\mathscr{L}_{\boldsymbol{x}}^{\phi_p}(\tilde{\boldsymbol{x}}). \tag{14}$$

Plugging the identity above into Equation 13 yields:

$$\boldsymbol{\nabla}(\mathscr{L}_{\boldsymbol{x}}^{\Phi} \circ F_{\boldsymbol{x}})(\boldsymbol{w}) = \frac{1}{P}\sum_{p=0}^{P-1}\left(\mathbf{J}_{F_{\boldsymbol{x}}}(\boldsymbol{w})^{\top}\boldsymbol{\nabla}\mathscr{L}_{\boldsymbol{x}}^{\phi_p}(\tilde{\boldsymbol{x}})\right)$$

$$= \mathbf{J}_{F_{\boldsymbol{x}}}(\boldsymbol{w})^{\top}\left(\frac{1}{P}\sum_{p=0}^{P-1}\boldsymbol{\nabla}\mathscr{L}_{\boldsymbol{x}}^{\phi_p}(\tilde{\boldsymbol{x}})\right), \tag{15}$$

where the latter equation holds by distributivity of matrix multiplication.

We now compute the expected value of $\boldsymbol{\nabla}\mathscr{L}_{\boldsymbol{x}}^{\phi_z}(\tilde{\boldsymbol{x}})$ for $z \sim \mathscr{U}_P$, i.e., a uniform distribution over $\mathscr{P}$:

$$\mathbb{E}_{z\sim\mathscr{U}_P}\left[\boldsymbol{\nabla}\mathscr{L}_{\boldsymbol{x}}^{\phi_z}(\tilde{\boldsymbol{x}})\right] = \frac{1}{P}\sum_{p=0}^{P-1}\boldsymbol{\nabla}\mathscr{L}_{\boldsymbol{x}}^{\phi_p}(\tilde{\boldsymbol{x}}). \tag{16}$$

We recognize the column vector on the right-hand side of Equation 15. Thus:

$$\boldsymbol{\nabla}(\mathscr{L}_{\boldsymbol{x}}^{\Phi} \circ F_{\boldsymbol{x}})(\boldsymbol{w}) = \mathbf{J}_{F_{\boldsymbol{x}}}(\boldsymbol{w})^{\top}\mathbb{E}_{z\sim\mathscr{U}_P}\left[\boldsymbol{\nabla}\mathscr{L}_{\boldsymbol{x}}^{\phi_z}(\tilde{\boldsymbol{x}})\right]$$

$$= \mathbb{E}_{z\sim\mathscr{U}_P}\left[\mathbf{J}_{F_{\boldsymbol{x}}}(\boldsymbol{w})^{\top}\boldsymbol{\nabla}\mathscr{L}_{\boldsymbol{x}}^{\phi_z}(\tilde{\boldsymbol{x}})\right], \tag{17}$$

where the latter equation holds by linearity of the expected value. Finally, we use the reverse form of the chain rule in Equation 14 to identify the expected SCRAPL gradient:

$$\boldsymbol{\nabla}(\mathscr{L}_{\boldsymbol{x}}^{\Phi} \circ F_{\boldsymbol{x}})(\boldsymbol{w}) = \mathbb{E}_{z\sim\mathscr{U}_P}\left[\boldsymbol{\nabla}(\mathscr{L}_{\boldsymbol{x}}^{\phi_z} \circ F_{\boldsymbol{x}})(\boldsymbol{w})\right] \tag{18}$$

concluding the proof.

## B    ADDITIONAL GRANULAR SYNTH EVALUATION RESULTS

Table 5: Loss function benchmark results for one optimization step (forward + backward, 32768 samples of audio, batch size 4, 1 thread, single precision, 1 NVIDIA RTX A5000 GPU, CUDA 12.4, PyTorch 2.8.0). SCRAPL paths are benchmarked individually and then aggregated across all paths using the median for time and interquartile range (IQR), and maximum for memory usage.

| Method | Median Time (ms) $\downarrow$ | IQR (ms) $\downarrow$ | Max. Memory (MB) $\downarrow$ |
|---|---|---|---|
| JTFS | 1730 | 23.9 | 12 967 |
| SCRAPL | 89.8 | 3.62 | 2503 |
| MSS Linear | 26.3 | 1.12 | 694 |
| MSS Log + Linear | 19.1 | 0.696 | 702 |
| MSS Revisited | **17.0** | **0.210** | **663** |
| MSS Random | 24.7 | 5.81 | 706 |
| MS-CLAP | 75.6 | 1.69 | 2032 |
| PANNs Wavegram-Logmel | 29.3 | 5.92 | 1360 |
| P-loss $(\dim(\boldsymbol{\theta}_{\text{synth}}) = 2)$ | 0.516 | 0.108 | 625 |

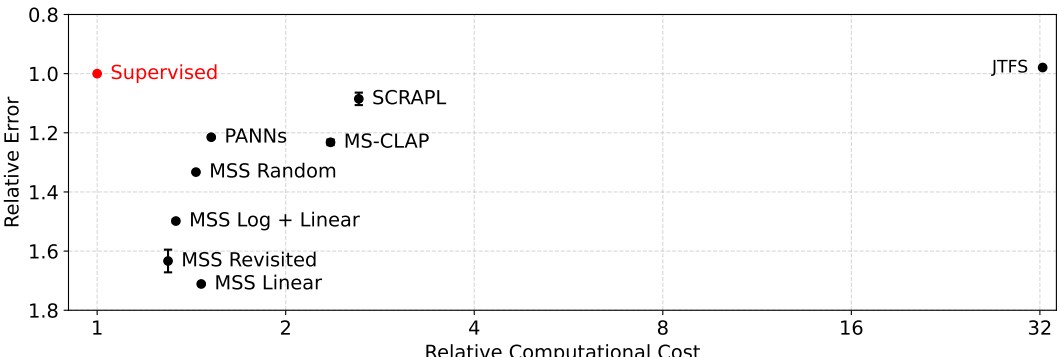

Figure 3:  Mean average JTFS perceptual audio distance (y-axis) versus computational cost (x-axis) of unsupervised sound matching models for the granular synthesis task. Both axes are rescaled by the performance of a supervised model with the same number of parameters. Whiskers denote 95% CI, estimated over 20 random seeds. Due to computational limitations, JTFS-based sound matching is evaluated only once.

Table 6: Statistical significance and relative improvement of each additional SCRAPL optimization technique for the unsupervised granular synth sound matching task. Convergence is defined as $\boldsymbol{\theta}_{\text{synth}} L_1 < 100$ ‰. See Table 2 for absolute results and comparisons to JTFS and P-loss. Uncertainties are 95% CI for 20 training runs using different random seeds. Due to computational limitations, the JTFS method is only evaluated once.

| Opt. | Test | Validation | |
|---|---|---|---|
| Technique | $\Delta\boldsymbol{\theta}_{\text{synth}} L_1$ ‰ $\downarrow$ | $\Delta$ Total Var. $\downarrow$ | $\Delta$ Conv. Steps $\downarrow$ |
| + $\mathscr{P}$-Adam | $-12.3 \pm 17$  $(p = 0.15)$ | $1.68 \pm 0.35$  $(p < 0.01)$ | $\mathbf{-2900} \pm 1800 \, (p < 0.01)$ |
| + $\mathscr{P}$-SAGA | $\mathbf{-13.6} \pm 5.6 \, (p < 0.01)$ | $\mathbf{-3.52} \pm 0.18$  $(p < 0.01)$ | $-710 \pm 450 \, (p < 0.01)$ |
| + $\theta$-IS | $-8.13 \pm 15$   $(p = 0.26)$ | $-0.188 \pm 0.092 \, (p < 0.01)$ | $-1280 \pm 410 \, (p < 0.01)$ |

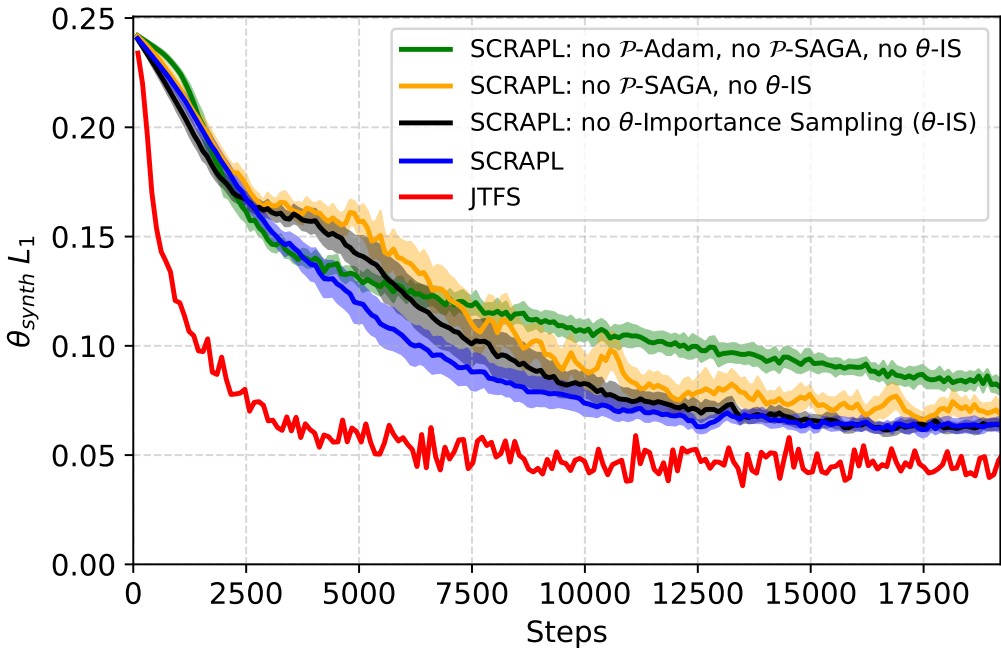

Figure 4: Validation convergence graphs of SCRAPL ablations and the JTFS for the unsupervised granular synth sound matching task. Shaded areas are 95% CI for 20 training runs using different random seeds. Due to computational limitations, the JTFS method is only evaluated once.

# C  ADDITIONAL CHIRPLET SYNTH EVALUATION RESULTS

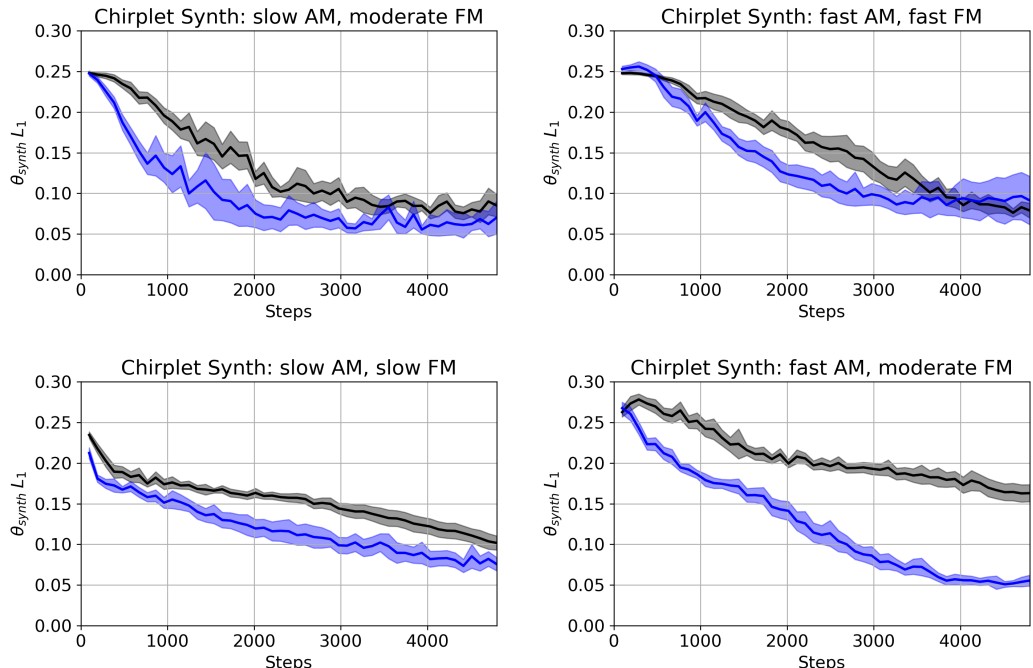

Figure 5: SCRAPL $\theta_{synth}$ $L_1$ validation values during training for four different AM / FM chirplet synths with two continuous $\theta_{synth}$ parameters: $\theta_{AM}$ and $\theta_{FM}$ (more details in Section 4.3). Blue is using the $\theta$-importance sampling initialization heuristic, and black is using uniform sampling. Shaded areas are 95% CI for 20 training runs using different random seeds.

Table 7: Convergence rate (CR) and steps for SCRAPL with and without the $\theta$-importance sampling initialization heuristic on unsupervised sound matching of four different AM / FM chirplet synths with two continuous $\theta_{synth}$ parameters: $\theta_{AM}$ and $\theta_{FM}$ (more details in Section 4.3). Convergence is defined as $L_1 < 100\,\%$ for $\theta_{AM}$ or $\theta_{FM}$. Uncertainties are 95% CI for 20 training runs using different random seeds.

| Sampling Method | Synth Configuration | | $\theta_{AM}$ | | $\theta_{FM}$ | |
|---|---|---|---|---|---|---|
| ($\pi$) | $\theta_{AM}$ (Hz) | $\theta_{FM}$ (oct/s) | CR ↑ | Conv. Steps ↓ | CR ↑ | Conv. Steps ↓ |
| Uniform | $1.0 - 2.0$ | $0.5 - 1.0$ | 60% | $3944 \pm 342$ | 45% | $4064 \pm 372$ |
| $\theta$-IS | | | **100%** | **$2002 \pm 324$** | **100%** | **$3134 \pm 492$** |
| Uniform | $1.0 - 2.0$ | $2.0 - 4.0$ | **100%** | $2203 \pm 135$ | **100%** | $1536 \pm 194$ |
| $\theta$-IS | | | **100%** | **$1099 \pm 173$** | **100%** | **$768 \pm 118$** |
| Uniform | $2.8 - 8.4$ | $2.0 - 4.0$ | 95% | $3254 \pm 250$ | 0% | N/A |
| $\theta$-IS | | | **100%** | **$1925 \pm 165$** | **100%** | **$2966 \pm 210$** |
| Uniform | $2.8 - 8.4$ | $4.0 - 12.0$ | **100%** | $3096 \pm 334$ | 95% | $3208 \pm 235$ |
| $\theta$-IS | | | 95% | **$2253 \pm 218$** | 95% | **$2178 \pm 173$** |

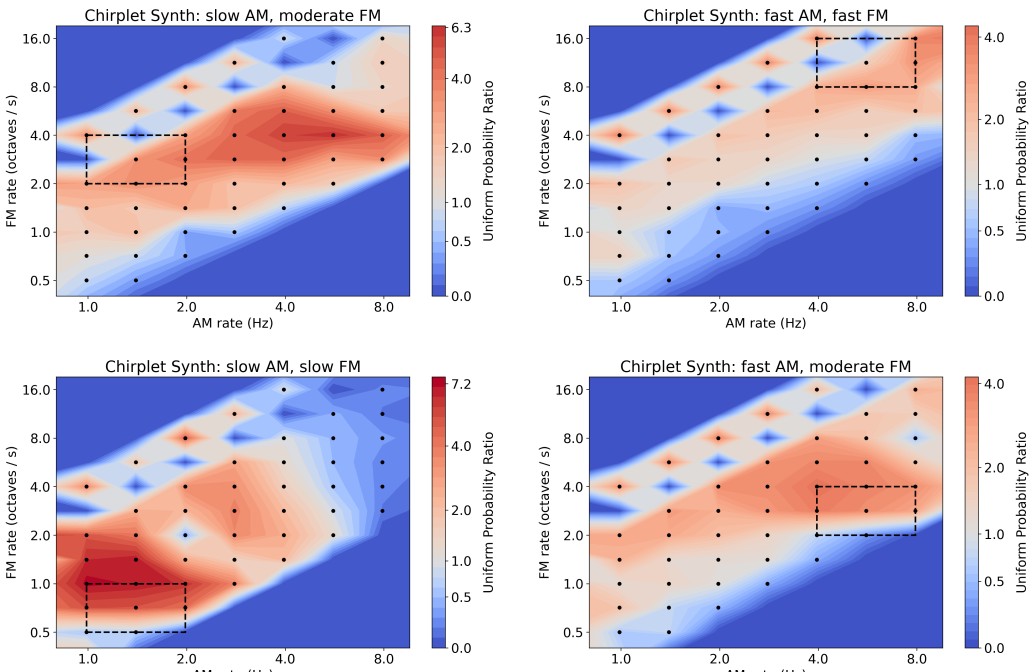

Figure 6: SCRAPL path $\theta$-importance sampling probabilities for four different AM / FM chirplet synths calculated from 1 batch of 32 log-uniformly randomly sampled $\theta_{\text{synth}}$ values. Black dots are individual path (wavelet) AM / FM center frequency locations and each dashed rectangle is the $\theta_{\text{synth}}$ range for each synth configuration. A uniform probability ratio of 1.0 means a path is sampled with probability $1/P$.

# D ADDITIONAL ROLAND TR-808 EVALUATION RESULTS

Table 8: Drum transient evaluation results for the unsupervised Roland TR-808 DDSP synth sound matching task with 14 continuous $\theta_{\text{synth}}$ parameters (more details in Section 4.4). Uncertainties are 95% CI for 40 training runs using different random seeds and dataset splits. Due to computational limitations, the JTFS method is only trained and evaluated for 4 random seeds.

| Method | Loudness $L_1 \downarrow$ | | Spectral Centroid $L_1 \downarrow$ | | Spectral Flatness $L_1 \downarrow$ | |
|---|---|---|---|---|---|---|
| | Micro | Meso | Micro | Meso | Micro | Meso |
| JTFS | **137** $\pm 10$ | **158** $\pm\ 20$ | **859** $\pm\ 68$ | **819** $\pm\ 96$ | 1200 $\pm\ 87$ | **1090** $\pm 110$ |
| SCRAPL | | | | | | |
| (no $\theta$-IS) | 389 $\pm 41$ | 460 $\pm\ 60$ | 1020 $\pm 100$ | 992 $\pm 110$ | 1800 $\pm 170$ | 1990 $\pm 180$ |
| SCRAPL | 374 $\pm 39$ | 377 $\pm\ 32$ | 1000 $\pm 120$ | 1080 $\pm 120$ | 1750 $\pm 270$ | 1820 $\pm 220$ |
| MSS Lin. | 381 $\pm 12$ | 2510 $\pm 480$ | 902 $\pm\ 27$ | 2350 $\pm 310$ | 962 $\pm\ 43$ | 3620 $\pm 680$ |
| MSS L+L | 492 $\pm 44$ | 1080 $\pm\ 91$ | 928 $\pm\ 65$ | 1380 $\pm\ 76$ | **916** $\pm\ 50$ | 1320 $\pm 150$ |
| MSS Rev. | 330 $\pm 21$ | 808 $\pm\ 40$ | 1070 $\pm\ 49$ | 1540 $\pm\ 53$ | 1390 $\pm\ 62$ | 2640 $\pm\ 96$ |
| MSS Rand. | 584 $\pm 75$ | 1030 $\pm\ 89$ | 1200 $\pm 100$ | 1350 $\pm\ 99$ | 1690 $\pm 120$ | 1950 $\pm 140$ |

Table 9: Drum decay evaluation results for the unsupervised Roland TR-808 DDSP synth sound matching task with 14 continuous $\theta_{\text{synth}}$ parameters (more details in Section 4.4). Uncertainties are 95% CI for 40 training runs using different random seeds and dataset splits. Due to computational limitations, the JTFS method is only trained and evaluated for 4 random seeds.

| Method | Loudness $L_1 \downarrow$ | | Spectral Centroid $L_1 \downarrow$ | | Spectral Flatness $L_1 \downarrow$ | |
|---|---|---|---|---|---|---|
| | Micro | Meso | Micro | Meso | Micro | Meso |
| JTFS | 315 $\pm\ 22$ | **355** $\pm 110$ | 614 $\pm\ 51$ | 617 $\pm\ 71$ | 527 $\pm\ 31$ | 718 $\pm 190$ |
| SCRAPL | | | | | | |
| (no $\theta$-IS) | 1810 $\pm 190$ | 2210 $\pm 210$ | 1530 $\pm 170$ | 1860 $\pm 170$ | 2620 $\pm 310$ | 3300 $\pm 370$ |
| SCRAPL | 1810 $\pm 160$ | 1740 $\pm 170$ | 1490 $\pm 120$ | 1470 $\pm 140$ | 2540 $\pm 290$ | 2480 $\pm 290$ |
| MSS Lin. | 357 $\pm\ 12$ | 1120 $\pm 260$ | 654 $\pm\ 18$ | 1110 $\pm 160$ | **472** $\pm\ 17$ | 1500 $\pm 350$ |
| MSS L+L | 389 $\pm\ 42$ | 466 $\pm\ 45$ | **563** $\pm\ 22$ | **597** $\pm\ 24$ | 565 $\pm\ 29$ | **644** $\pm\ 51$ |
| MSS Rev. | **279** $\pm\ 12$ | 494 $\pm\ 22$ | 589 $\pm\ 21$ | 801 $\pm\ 29$ | 552 $\pm\ 22$ | 846 $\pm\ 29$ |
| MSS Rand. | 453 $\pm\ 21$ | 485 $\pm\ 24$ | 660 $\pm\ 27$ | 640 $\pm\ 35$ | 594 $\pm\ 30$ | 658 $\pm\ 33$ |

# E    EXPERIMENT TRAINING DETAILS AND HYPERPARAMETERS

Table 10: Unsupervised granular synth sound matching task hyperparameters.

| Category | Hyperparameter Name | Value |
|---|---|---|
| Data | $N$ (# of examples) | 5120 |
|  | train / val / test split | 60% / 20% / 20% |
| Encoder | # of parameters | 604 K |
|  | CQT # of octaves | 5 |
|  | CQT bins / octave | 12 |
|  | CQT hop length | 256 |
|  | CQT postprocessing | `log1p` |
|  | CNN # of conv. blocks | 5 |
|  | CNN kernel size | $5 \times (3, 3)$ |
|  | CNN stride | $5 \times (1, 1)$ |
|  | CNN pooling | $5 \times (2, 2)$ |
|  | CNN conv. block channels | 128 |
|  | CNN activation function | `PReLU` |
|  | CNN embedding dim. | 64 |
|  | CNN dense layer dropout prob. | 0.5 |
| Decoder (Synth) | $\dim(\boldsymbol{\theta}_{\text{synth}})$ | 2 |
|  | sampling rate | 8192 Hz |
|  | $T$ (# of samples) | 32768 |
|  | max. # of grains | 64 |
|  | grain # of samples | 4096 |
|  | min. grain pitch | 256 Hz |
|  | max. grain pitch | 2048 Hz |
| SCRAPL & JTFS | $J$ | 12 |
|  | $Q_1$ | 8 |
|  | $Q_2$ | 2 |
|  | $J_{\text{fr}}$ | 3 |
|  | $Q_{\text{fr}}$ | 2 |
|  | $T$ | 4096 |
|  | $F$ | 8 |
|  | $\rho$ | identity function |
|  | $P$ (# of paths) | 315 |
| $\theta$-Importance Sampling | $N_{\text{IS}}$ (# of examples) | 320 |
|  | max. # of deflated power iterations | 20 |
| Training | # of random seed training runs | 20 |
|  | epochs | 200 |
|  | batch size | 32 |
|  | starting learning rate | $1 \times 10^{-5}$ |
|  | learning rate scheduler | none |
|  | Adam $\beta_1$ | 0.9 |
|  | Adam $\beta_2$ | 0.999 |
|  | weight decay | 0.01 |

Table 11: Unsupervised AM / FM chirplet synth sound matching task hyperparameters.

| Category | Hyperparameter Name | Value |
|---|---|---|
| Data | $N$ (# of examples) | 5120 |
| | train / val / test split | 60% / 20% / 20% |
| Encoder | # of parameters | 604 K |
| | CQT # of octaves | 5 |
| | CQT bins / octave | 12 |
| | CQT hop length | 256 |
| | CQT postprocessing | log1p |
| | CNN # of conv. blocks | 5 |
| | CNN kernel size | $5 \times (3, 3)$ |
| | CNN stride | $5 \times (1, 1)$ |
| | CNN pooling | $5 \times (2, 2)$ |
| | CNN conv. block channels | 128 |
| | CNN activation function | PReLU |
| | CNN embedding dim. | 64 |
| | CNN dense layer dropout prob. | 0.5 |
| Decoder (Synth) | $\dim(\boldsymbol{\theta}_{\text{synth}})$ | 2 |
| | sampling rate | 8192 Hz |
| | $T$ (# of samples) | 32768 |
| | chirplet center frequency | 512 Hz |
| | chirplet bandwidth | 2 octaves |
| | min. time shift | -2048 samples |
| | max. time shift | +2048 samples |
| SCRAPL & JTFS | $J$ | 12 |
| | $Q_1$ | 8 |
| | $Q_2$ | 2 |
| | $J_{\text{fr}}$ | 3 |
| | $Q_{\text{fr}}$ | 2 |
| | $T$ | 4096 |
| | $F$ | 8 |
| | $\rho$ | identity function |
| | $P$ (# of paths) | 315 |
| $\theta$-Importance Sampling | $N_{\text{IS}}$ (# of examples) | 32 |
| | max. # of deflated power iterations | 20 |
| Training | # of random seed training runs | 20 |
| | epochs | 50 |
| | batch size | 32 |
| | starting learning rate | $1 \times 10^{-4}$ |
| | learning rate scheduler | none |
| | Adam $\beta_1$ | 0.9 |
| | Adam $\beta_2$ | 0.999 |
| | weight decay | 0.01 |

Table 12: Unsupervised Roland TR-808 synth sound matching task hyperparameters.

| Category | Hyperparameter Name | Value |
|---|---|---|
| Data | $N$ (# of examples) | 681 |
| | $N_{\text{bass drum}}$ | 215 |
| | $N_{\text{snare}}$ | 240 |
| | $N_{\text{tom}}$ | 189 |
| | $N_{\text{hi-hat}}$ | 37 |
| | $N_{\text{train}}$ | 425 |
| | $N_{\text{val}}$ | 128 |
| | $N_{\text{test}}$ | 128 |
| Encoder | # of parameters | 724 K |
| | CQT # of octaves | 9 |
| | CQT bins / octave | 12 |
| | CQT hop length | 256 |
| | CQT postprocessing | `log1p` |
| | CNN # of conv. blocks | 5 |
| | CNN kernel size | $5 \times (3, 3)$ |
| | CNN stride | $5 \times (1, 1)$ |
| | CNN pooling | $(2, 2), (2, 2), (2, 3), (3, 3), (3, 3)$ |
| | CNN conv. block channels | 128 |
| | CNN activation function | `PReLU` |
| | CNN embedding dim. | 128 |
| | CNN dense layer dropout prob. | 0.25 |
| Decoder (Synth) | $\dim(\boldsymbol{\theta}_{\text{synth}})$ | 14 |
| | sampling rate | 44100 Hz |
| | $T$ (# of samples) | 44100 |
| | min. time shift | -2048 samples |
| | max. time shift | +2048 samples |
| SCRAPL & JTFS | $J$ | 12 |
| | $Q_1$ | 8 |
| | $Q_2$ | 2 |
| | $J_{\text{fr}}$ | 5 |
| | $Q_{\text{fr}}$ | 2 |
| | $T$ | 2048 |
| | $F$ | 1 |
| | $\rho$ | `log1p` |
| | $P$ (# of paths) | 483 |
| $\theta$-Importance Sampling | $N_{\text{IS}}$ (# of examples) | 16 |
| | max. # of deflated power iterations | 20 |
| Training | # of random seed training runs | 40 |
| | epochs | 50 |
| | batch size | 8 |
| | starting learning rate | $1 \times 10^{-4}$ |
| | learning rate scheduler | linearly decreasing until $1 \times 10^{-5}$ |
| | Adam $\beta_1$ | 0.9 |
| | Adam $\beta_2$ | 0.999 |
| | weight decay | 0.01 |

