# OpenReview forum: "SCRAPL: Scattering Transform with Random Paths for Machine Learning"
_ICLR.cc/2026/Conference — ICLR 2026 Poster_

### Official Review · Reviewer_3oJ4 · 2025-10-30

**Soundness:** 2
**Presentation:** 3
**Contribution:** 2
**Rating:** 4
**Confidence:** 4

**Summary:**

This article introduces an efficient way to compute  the wavelet scattering transform based on random sampling. This allows to learn auto-encoder at scale To measure dissimilarity between isolated sounds, under a perceptual distance defined by this transform. To avoid the variance of the random sampling, various methods from the literature of stochastic gradient descent (SGD) are considered.  The proposed scheme is applied to unsupervised sound matching, showing comparable performance compared to joint-time frequency scattering transform and other baseline methods.

**Strengths:**

-	The problem on how to compute the scattering transform efficiently is well motivated.
-	The obtained results are very promising.
-	The article is mostly well written.

**Weaknesses:**

- The modified SGD in P-Adam (eq 8) and P-SAGA (eq 9) seem to be quite heuristic as there is no convergence analysis in the literature. For example, in P-SAGA, which rely on the gradients from P-ADAM is not a standard SAGA method given in the citation (Defazio et al 2014).
- The section 3.4 is not so clear and might contain some technical flaw.
- It is not clear whether the proposed scheme has any numerical convergence.
- The evaluation metric is not so clear in Table 1 to non-expert. In Table 4, the evaluation scores seem to be biased towards the loss function that you are optimizing. Any human-based or more objective evaluation would be needed to compare different methods.

**Questions:**

- Is it possible to add some reference to the convergence analysis of P-Adam and P-SAGA methods? Why don’t you just use the standard SAGA method?
- In Section 3.4, is theta belong R^U according to eq. 11? Is theta = the output of encoder?
- how pi_p in eq. 10 is used next?
- Is C_{u,p} in eq. 12 always positive? If yes, can you explain clearly? If not , how to ensure that pi_p is a probability in eq. 10?
- To understand the numerical convergence of SCRARL, an analysis on the final training loss (eq. 4) is needed to compare with JTFS.
- Could you give a definition on the theta_syn, theta_density etc to explain Table 1?
- How the supervised version P-loss method works? Why do you choose this to compare with the other methods?

---

> ### Author Response · Authors · 2025-11-20
> **Author Response 1**
>
> Thank you for the feedback, we address each comment below. We will post another comment once the updated paper draft with the incorporated feedback has been uploaded.
>
> *The modified SGD in P-Adam (eq 8) and P-SAGA (eq 9) seem to be quite heuristic as there is no convergence analysis in the literature. For example, in P-SAGA, which rely on the gradients from P-ADAM is not a standard SAGA method given in the citation (Defazio et al 2014).
> It is not clear whether the proposed scheme has any numerical convergence.
> Is it possible to add some reference to the convergence analysis of P-Adam and P-SAGA methods?*
>
> We agree that P-Adam and P-SAGA are heuristics with no convergence guarantees, especially for our experiments which are all non-convex. However, the extremely popular Adam algorithm also has no general convergence guarantees, even for convex problems [1, 2]. A full convergence analysis of P-Adam or P-SAGA or the combination of both would be considered future work and could be a separate publication, as is exhibited by the large number of papers analyzing the convergence of optimizers like Adam or variance reduction algorithms like SAGA. We emphasize that our statistically significant ablations in Table 2 empirically indicate that these algorithms improve convergence in practice and that our proof in Appendix A that SCRAPL is unbiased is a fundamental first step for proving stochastic optimization convergence. Finally, we will review the relevant literature on Adam and SAGA convergence and provide additional context in the paper about their convergence and how this potentially relates to P-Adam and P-SAGA.
>
> [1] Reddi, S. J., Kale, S., & Kumar, S. (2018). On the Convergence of Adam and Beyond. In Proceedings of the International Conference on Learning Representations. Retrieved from https://openreview.net/forum?id=ryQu7f-RZ
>
> [2] Kim, G. Y., & Oh, M.-H. (2025). ADAM Optimization with Adaptive Batch Selection. In Proceedings of the International Conference on Learning Representations. Retrieved from https://openreview.net/forum?id=BZrSCv2SBq
>
> *The section 3.4 is not so clear and might contain some technical flaw.*
>
> We have edited and improved the clarity of Section 3.4 and have carefully checked it for consistency and rigor.
>
> *The evaluation metric is not so clear in Table 1 to non-expert.*
>
> We have included additional context in the caption of Table 1 to make it more understandable for non-experts.
>
> *In Table 4, the evaluation scores seem to be biased towards the loss function that you are optimizing. Any human-based or more objective evaluation would be needed to compare different methods.*
>
> We are looking into re-running the experiments with an additional perceptual evaluation metric like FAD and will update the paper accordingly. We will post another comment once we have done so, since this will take some time to complete. We also clarify in the paper that the JTFS has been found to correlate with human perception [3] and is therefore still a reasonable evaluation metric to use.
>
> [3] Lostanlen, V., El-Hajj, C., Rossignol, M., Lafay, G., Andén, J., & Lagrange, M. (2021). Time-Frequency Scattering Accurately Models Auditory Similarities Between Instrumental Playing Techniques. EURASIP Journal on Audio, Speech, and Music Processing, 1(1), 1–21. doi:10.1186/s13636-020-00187-z
>
> *Why don’t you just use the standard SAGA method?*
>
> Unfortunately, outside of toy examples, the standard SAGA algorithm is prohibitively expensive for neural networks since it requires keeping a copy of all gradients in memory for every sample in the dataset. In contrast to this, P-SAGA only requires P copies of the gradients, of which only one is loaded in the GPU’s VRAM at any point in time since SCRAPL is stochastically sampling only one path each step. This makes it ~10-100x less memory intensive for our experiments and therefore feasible for real-world applications. We have clarified this in the paper.
>
> *In Section 3.4, is theta belong R^U according to eq. 11? Is theta = the output of encoder?
> how pi_p in eq. 10 is used next?*
>
> Yes, the dimension of $\theta$ is U ($\theta_{u=0}^{U-1}$) and it is the output of the encoder which is then fed to the decoder (synth) to generate sound. $\pi_{p}$ is used to sample paths non-uniformly in the SCRAPL algorithm written out in Appendix B. We have edited and improved the clarity of Section 3.4.
>
> *Is C_{u,p} in eq. 12 always positive? If yes, can you explain clearly? If not , how to ensure that pi_p is a probability in eq. 10?*
>
> Yes, $C_{u,p}$ in eq. 12 is always positive because we use the magnitude of the largest eigenvalue. Since we take inspiration from the procedure outlined by Schmidt et al. (2017), we are estimating the Lipschitz constant (which by definition is always positive) using the largest eigenvalue of the Hessian, which means only the magnitude is of interest to us. We have clarified this in the paper.

---

> > ### Author Response · Authors · 2025-11-20
> > **Author Response 2**
> >
> > *To understand the numerical convergence of SCRAPL, an analysis on the final training loss (eq. 4) is needed to compare with JTFS.*
> >
> > We are looking into re-running the experiments to create another figure similar to Figure 1, but this time using the Euclidean JTFS distance between the input and reconstructed audio. This will enable a direct comparison between SCRAPL / other methods and the JTFS. However, this will take a lot of time and may not be complete by the rebuttal deadline, we will post another comment here accordingly.
> >
> > *Could you give a definition on the theta_syn, theta_density etc to explain Table 1?*
> >
> > For improved clarity, we have updated the caption of Table 1 with the explanation of the granular synth parameters provided in Section 4.2.
> >
> > *How the supervised version P-loss method works?*
> >
> > The supervised P-Loss baseline involves training just the encoder (no differentiable decoder / synthesizer) with mean squared error loss to predict the synth parameters ($\theta_{u=0}^{U-1}$) directly. This is only possible if the ground truth synth parameters are known which is often not the case in sound matching and perceptual quality assessment tasks (hence the emphasis on unsupervised experiments). We have explained P-Loss in more detail in the paper.
> >
> > *Why do you choose this to compare with the other methods?*
> >
> > We chose to compare against supervised P-Loss because it provides an upper bound on performance given a specific neural network architecture, dataset, input audio representation, and hyperparameters. In the related literature [4] it is common to compare against P-Loss for unsupervised sound matching tasks when the ground truth labels are available. We have clarified this in the paper.
> >
> > [4] Hayes, B., Shier, J., Fazekas, G., McPherson, A., & Saitis, C. (2024). A review of differentiable digital signal processing for music and speech synthesis. Frontiers in Signal Processing, 3–2023. doi:10.3389/frsip.2023.1284100

---

> > > ### Comment · Reviewer_3oJ4 · 2025-11-26
> > > **futher discussion**
> > >
> > > The following points still require some clarification:
> > > - Is it possible to add some reference to the convergence analysis of P-Adam and P-SAGA methods? Why don’t you just use the standard SAGA method?
> > > - To understand the numerical convergence of SCRARL, an analysis on the final training loss (eq. 4) is needed to compare with JTFS.
> > >
> > > I am not suggesting to redo your Figure 1, but rather to analyze the results that you already should have. I am wondering the optimization convergence property of SCRARL to optimize your training loss in (4) since there is a performance gap between SCRARL and JTFS in your results (table 1 and 4). Is it due to the fact that the proposed P-Adam or P-SAGA does not optimize well the training loss? If I understood, when you optimize directly (4), you obtained the JTFS results, isn't it?
> > >
> > > - Is C_{u,p} in eq. 12 always positive? If yes, can you explain clearly? If not , how to ensure that pi_p is a probability in eq. 10?
> > >
> > > It is not clear to me why there is a Hessian matrix in eq 12, which is symmetric? I thought that you have a non-symmetric matrix M here to compute its lambda_max. The sentence on line 236 does not make sense: "where λmax(M) is the largest eigenvalue of a square matrix M;" unless M is symmetric, or say the largest amplitude if M is non-symmtric?
> > >
> > >
> > > Best,
> > > reviewer

---

> > > > ### Author Response · Authors · 2025-11-28
> > > > **Author Response 3**
> > > >
> > > > Thank you for your quick response, we address each comment below. We will post another comment once the updated paper draft with the incorporated feedback has been uploaded.
> > > >
> > > > *The following points still require some clarification:*
> > > >
> > > > - *Is it possible to add some reference to the convergence analysis of P-Adam and P-SAGA methods? Why don’t you just use the standard SAGA method?*
> > > > - *To understand the numerical convergence of SCRARL, an analysis on the final training loss (eq. 4) is needed to compare with JTFS.*
> > > >
> > > > *I am not suggesting to redo your Figure 1, but rather to analyze the results that you already should have. I am wondering the optimization convergence property of SCRARL to optimize your training loss in (4) since there is a performance gap between SCRARL and JTFS in your results (table 1 and 4). Is it due to the fact that the proposed P-Adam or P-SAGA does not optimize well the training loss? If I understood, when you optimize directly (4), you obtained the JTFS results, isn't it?*
> > > >
> > > > Thank you for clarifying your initial questions about convergence and yes, the JTFS results are obtained from directly optimizing Equation 4.
> > > >
> > > > *Why don’t you just use the standard SAGA method? Is it possible to add some reference to the convergence analysis of P-Adam and P-SAGA methods?*
> > > >
> > > > Please see our previous response (Author Response 1) for why the standard SAGA method cannot be used outside of toy examples as well as a discussion on additional references for a convergence analysis of P-Adam and P-SAGA. We also provide more information related to this below.
> > > >
> > > > *To understand the numerical convergence of SCRARL, an analysis on the final training loss (eq. 4) is needed to compare with JTFS.*
> > > >
> > > > Indeed, from Table 1 and Figure 1, we see that SCRAPL does not converge as well as the JTFS (i.e. standard full-tree scattering transforms). Intuitively, this is to be expected since full-tree scattering evaluates all paths at every step and therefore provides P times more information and extra computation per step compared to SCRAPL, and crucially, both are being trained for the *same* number of steps in our experiments. The fact that SCRAPL can come close to full-tree scattering in terms of accuracy, but with orders of magnitude less computation is one of our main contributions. From our proof in Appendix A of Proposition 3.1 that “vanilla” SCRAPL is an unbiased estimator of full-tree scattering transforms, we would expect “vanilla” SCRAPL to converge to the same result as full-tree scattering after P times more training steps using the stochastic gradient descent algorithm applied to a convex optimization task.
> > > >
> > > > However, in practice, more advanced optimization algorithms like Adam are required for many real-world, non-convex problems, like our experiments. Additionally, since P is generally in the order of hundreds of paths, conducting experiments where SCRAPL is trained P times longer is computationally very expensive. While P-Adam, P-SAGA, and $\theta$-IS reduce the variance and significantly improve the convergence of “vanilla” SCRAPL and optimization of the training loss, which we empirically demonstrate in Table 2,  they are unable to improve it all the way to the same performance of full-tree scattering transforms (and this would be a remarkable result if they could).
> > > >
> > > > We have provided additional discussion in the paper in Section 5.1 and further emphasized the multiple orders of magnitude reduced computation of the SCRAPL algorithm compared to full-tree scattering.
> > > >
> > > > *It is not clear to me why there is a Hessian matrix in eq 12, which is symmetric? I thought that you have a non-symmetric matrix M here to compute its lambda_max. The sentence on line 236 does not make sense: "where λmax(M) is the largest eigenvalue of a square matrix M;" unless M is symmetric, or say the largest amplitude if M is non-symmtric?*
> > > >
> > > > Matrix M in our setup is necessarily square and symmetric. This becomes clear from a dimensionality analysis of equations 11 and 12. We see that $s_{\boldsymbol{x},u,p}$ is a scalar and $E_{\boldsymbol{x},u}(\boldsymbol{w})$ returns a scalar: the decoder (synth) parameter $\theta_u$. Therefore the gradient of the encoder w.r.t the weights $\boldsymbol{\nabla}E_{\boldsymbol{x},u}(\boldsymbol{w})$ is a row vector of dimension $w$. Multiplying by the scalar $s_{\boldsymbol{x},u,p}$ and then taking the gradient again w.r.t. the weights results in a Hessian square matrix of size $w \times w$ and is guaranteed to be symmetric. This is due to Clairaut’s theorem which intuitively states that the order of the two partial derivatives does not matter. We have clarified this in the paper and also revised the bold notation for vector values for the sake of consistency.

---

> > > > > ### Author Response · Authors · 2025-12-03
> > > > > **Rebuttal Revision**
> > > > >
> > > > > We have uploaded the rebuttal revision of the paper, which incorporates the feedback from all four reviewers.

---

### Official Review · Reviewer_iLNX · 2025-11-01

**Soundness:** 2
**Presentation:** 3
**Contribution:** 3
**Rating:** 6
**Confidence:** 3

**Summary:**

This paper presents a computationally efficient stochastic approximation to the full scattering transform to enable use of scattering based similarity as a loss function, one that relates to perceptual quality, in generative modeling.  The proposed approach consists of four steps: a) A sampling of scattering “paths”, b) a path-wise adaptive momentum estimation and using that in an extension of the Adam optimization algorithm, c) path-wise stochastic average gradient with acceleration using previous gradient values, and d) an importance sampling approach to bring stochastic approximation of spectral loss closer to P-loss.  Empirical results are presented on unsupervised sound matching experiments with granular synth, chirplet synth, and Roland TR-808 synth.

**Strengths:**

* A stochastic approach that makes optimization under scattering transforms computationally efficient
* Empirical results demonstrating that proposed approach achieves accuracies within a factor of two relative to the extensive joint time-frequency scattering transforms, while being within 3x of the much more efficient multi-scale spectral loss approach.
* Clear, well written paper

**Weaknesses:**

* The empirical results on a sound matching setup which is of relatively limited interest to the broader NeurIPS and ML community.  Results on more audio generation tasks utilizing perceptual qualities of scattering transforms will strengthen the paper significantly.

**Questions:**

n/a

---

> ### Author Response · Authors · 2025-11-20
> **Author Response 1**
>
> Thank you for the feedback, we address each comment below. We will post another comment once the updated paper draft with the incorporated feedback has been uploaded.
>
> *The empirical results on a sound matching setup which is of relatively limited interest to the broader NeurIPS and ML community. Results on more audio generation tasks utilizing perceptual qualities of scattering transforms will strengthen the paper significantly.*
>
> Due to time constraints, we have not tested SCRAPL on more audio tasks. However, any generative audio task that contains spectrotemporal modulations or both transients and sustained components like music, speech, and animal vocalizations would be well suited for evaluation using SCRAPL and the JTFS. While we do not foresee any challenges extending it to other audio tasks like speech enhancement and automatic mixing, we have clarified in the paper that the SCRAPL algorithm is a general algorithm that applies to all scattering transforms with depth-first search and we suggest additional audio and vision tasks as potential directions for future work.

---

> > ### Author Response · Authors · 2025-12-03
> > **Rebuttal Revision**
> >
> > We have uploaded the rebuttal revision of the paper, which incorporates the feedback from all four reviewers.

---

### Official Review · Reviewer_7Spt · 2025-11-01

**Soundness:** 4
**Presentation:** 4
**Contribution:** 4
**Rating:** 8
**Confidence:** 4

**Summary:**

This paper introduces SCRAPL (Scattering transform with Random Paths for machine Learning), a stochastic optimization framework that accelerates the use of multivariable scattering transforms as differentiable loss functions in deep learning. Scattering transforms (ST) are wavelet-based operators offering invariance and stability properties useful for perceptual modeling, but their computational cost has hindered large-scale training. SCRAPL addresses this by sampling a subset of ST “paths” at each iteration, yielding unbiased stochastic gradient estimates. To reduce variance and improve convergence, the authors introduce P-Adam (path-wise adaptive moment estimation) and P-SAGA (path-wise stochastic average gradient) algorithms, as well as a θ-importance sampling heuristic that biases path sampling toward perceptually informative regions. The method is implemented for the joint time–frequency scattering transform (JTFS) and evaluated on three differentiable digital signal processing (DDSP) tasks: granular synth, chirplet synth, and Roland TR-808 sound matching. Results show SCRAPL achieves accuracy close to JTFS at a fraction of its computational cost, outperforming existing perceptual losses such as multiscale spectral loss (MSS) and audio embedding distances.

**Strengths:**

SCRAPL is a novel and technically well-motivated approach that makes scattering transforms practical for end-to-end differentiable learning. The paper is clearly written and grounded in both theory and application. The proposed stochastic approximation is theoretically justified (unbiased gradient under uniform sampling) and supported by well-designed optimization variants (P-Adam, P-SAGA) that address variance and non-i.i.d. gradient issues. The θ-importance sampling heuristic is a creative contribution that connects the optimization strategy to perceptual signal characteristics. Experimental validation is thorough, spanning synthetic and real-world DDSP tasks, with convincing quantitative and qualitative results. The work provides clear computational benchmarks, reproducibility details, and a public implementation, all of which increase its credibility and impact. Overall, the paper bridges an important gap between scattering theory and modern deep learning practice.

**Weaknesses:**

While the proposed framework is promising, certain limitations remain. The experiments are largely confined to audio applications, and it is unclear how well SCRAPL generalizes to image or other modalities that use scattering transforms. Although θ-importance sampling improves convergence, its computation involves complex gradient–Hessian interactions that may limit scalability in higher-dimensional settings. The empirical comparisons, while comprehensive, rely on relatively small neural networks and do not explore how SCRAPL performs in large-scale or supervised training contexts. The ablation studies, though informative, could include statistical significance tests to strengthen claims. Finally, the paper’s reliance on many heuristic design choices (e.g., importance weighting, hyperparameter schedules) makes it difficult to assess robustness across diverse architectures and datasets.

**Questions:**

Have you tested or considered SCRAPL in vision tasks using 2D scattering transforms (e.g., roto-translation ST)? If not, do you anticipate any challenges extending it to images?

Could you clarify how often this importance distribution must be recomputed during training, and how its computational cost compares to one full scattering transform?

---

> ### Author Response · Authors · 2025-11-20
> **Author Response 1**
>
> Thank you for the feedback, we address each comment below. We will post another comment once the updated paper draft with the incorporated feedback has been uploaded.
>
> *The experiments are largely confined to audio applications, and it is unclear how well SCRAPL generalizes to image or other modalities that use scattering transforms.*
>
> Due to time constraints, we have not tested SCRAPL on other modalities like images. While we do not foresee any challenges extending it to other modalities, we have clarified in the paper that the SCRAPL algorithm is a general algorithm that applies to all scattering transforms with depth-first search and we suggest additional audio and vision tasks like adversarial image generation and texture synthesis as potential directions for future work.
>
> *Although θ-importance sampling improves convergence, its computation involves complex gradient–Hessian interactions that may limit scalability in higher-dimensional settings.*
>
> The computation required for $\theta$-IS can be trivially parallelized across both $\mathcal{P}$ and $\theta_{u=0}^{U-1}$ which we take advantage of in our experiments. We have clarified this in the paper. We also write that analyzing the behavior of $\theta$-IS for higher-dimensional synth parameters or ones with complex interactions is an area we are actively investigating as future work. Finally, we emphasize that even without $\theta$-IS, SCRAPL still enables a new class of problems to be optimized in a computationally tractable way.
>
> *The empirical comparisons, while comprehensive, rely on relatively small neural networks and do not explore how SCRAPL performs in large-scale or supervised training contexts.*
>
> We have mentioned in the paper that experiments on large-scale or supervised training contexts are compelling directions for future work. We also emphasize that for perceptual quality assessment tasks like sound matching where the decoder is an interpretable synth, this additional inductive bias makes relatively small neural networks for the encoder a common implementation choice [1]. Additionally, related work [2] shows that perceptual sound matching for percussion does not necessarily benefit from larger models, whereas changing the optimization algorithm can improve convergence significantly.
>
> [1] Hayes, B., Shier, J., Fazekas, G., McPherson, A., & Saitis, C. (2024). A review of differentiable digital signal processing for music and speech synthesis. Frontiers in Signal Processing, 3–2023. doi:10.3389/frsip.2023.1284100
>
> [2] Han, H., Lostanlen, V., & Lagrange, M. (2025, September). Gradient Clipping Improves Neural Network Optimization for Perceptual Sound Matching. Proceedings of the European Signal Processing Conference (EUSIPCO). Retrieved from https://hal.science/hal-05124224
>
> *The ablation studies, though informative, could include statistical significance tests to strengthen claims.*
>
> We have added information about statistical significance to the ablation Table 2 in addition to the existing 95% confidence intervals.
>
> *Finally, the paper’s reliance on many heuristic design choices (e.g., importance weighting, hyperparameter schedules) makes it difficult to assess robustness across diverse architectures and datasets.*
>
> There are no hyperparameter schedules mentioned or used in the paper besides very standard linear decay of the learning rate. While P-Adam, P-SAGA, and $\theta$-IS indeed increase the complexity of SCRAPL, P-Adam and P-SAGA introduce no additional hyperparameters over “vanilla” Adam. $\theta$-IS introduces two new hyperparameters: how many data samples to use and the maximum number of deflated power iterations. Both did not require any tuning for all our experiments and were chosen heuristically based on available computing power. We have clarified this in the paper. Additionally, we emphasize that even without these three extra algorithms, stochastic approximation of scattering transforms (“vanilla” SCRAPL) still enables a new class of problems to be optimized and outperforms all other methods as shown in Table 2 and Figure 2.

---

> > ### Author Response · Authors · 2025-11-20
> > **Author Response 2**
> >
> > *Have you tested or considered SCRAPL in vision tasks using 2D scattering transforms (e.g., roto-translation ST)? If not, do you anticipate any challenges extending it to images?*
> >
> > Due to time constraints, we have not tested SCRAPL on any vision tasks. While we do not foresee any challenges extending it to images, we have clarified in the paper that the SCRAPL algorithm is a general algorithm that applies to all scattering transforms with depth-first search and we suggest additional audio and vision tasks like adversarial image generation and texture synthesis as potential directions for future work.
> >
> > *Could you clarify how often this importance distribution must be recomputed during training, and how its computational cost compares to one full scattering transform?*
> >
> > The $\theta$-IS sampling probability distribution ($\pi_{p}$) is computed once at the beginning before training on $N_{\text{IS}}$ datapoints for each decoder synth parameter ($\theta_{u=0}^{U-1}$) and path $\mathcal{P}$. This means it is approximately equal to $N_{\text{IS}} \times U$ times the cost of one full scattering transform across all $\mathcal{P}$ with some small extra overhead. Crucially, the computation required for $\theta$-IS can be trivially parallelized across both $\mathcal{P}$ and $\theta_{u=0}^{U-1}$ which we take advantage of in our experiments. We have clarified this in the paper.

---

> > > ### Author Response · Authors · 2025-12-03
> > > **Rebuttal Revision**
> > >
> > > We have uploaded the rebuttal revision of the paper, which incorporates the feedback from all four reviewers.

---

### Official Review · Reviewer_HNRU · 2025-11-02

**Soundness:** 1
**Presentation:** 3
**Contribution:** 3
**Rating:** 6
**Confidence:** 4

**Summary:**

The paper introduces a stochastic optimization scheme to make the computationally expensive wavelet scattering transform (ST) viable for large-scale machine learning, particularly in Differentiable Digital Signal Processing (DDSP). The core problem addressed is that the full scattering transform, the Joint Time-Frequency Scattering Transform (JTFS), may be used as a perceptually relevant loss function for tasks like sound matching, but its complexity makes it too slow and memory-intensive for use in stochastic gradient descent. The proposed solution, SCRAPL (Scattering transform with Random Paths for machine Learning), uses a stochastic approach to efficiently approximate the gradient of the full ST loss.
The study reports the results for using SCRAPL in the context unsupervised sound matching—a nonlinear inverse problem where an autoencoder learns the control parameters of an audio synthesizer (DDSP)—for which SCRAPL demonstrates a superior tradeoff between accuracy and computational efficiency compared to other losses.

**Strengths:**

•	SCRAPL makes the Joint Time-Frequency Scattering (JTFS) transform feasible for use in the loss function of large-scale deep learning by overcoming its prohibitively high computational and memory costs.
•	The paper proposes an innovative stochastic approximation scheme—sampling just a single path—which guarantees an unbiased estimate of the true loss gradient in expectation, together with a novel Path-Wise Optimization (P-Adam) to manage the high variance of the single-path sampling, stabilizing convergence by maintaining individual moment statistics for every scattering path.
•	The experimental evaluation is performed in the context of unsupervised sound matching with DDSP. It demonstrates that despite the stochastic approximation, the proposed algorithm offers sound matching results that are close to the full JTFS, while achieving a significantly smaller memory footprint, proving its practical efficiency.

**Weaknesses:**

•	Limited Experimental Scope and Generalizability: The evaluation is performed on a relatively limited number of sound matching examples, which does not permit concluding on the general applicability of the method across different audio domains or tasks.
•	When listening to the audio examples, one should note that while the sound matching algorithm converges, the perceptual similarity of the original and generated sounds is not generally satisfying. In quite a few of the examples, the output sound is perceptually very far from the target. This calls the usefulness of the JTFS, and with it the SCRAPL algorithm, into question.
•	Increased Implementation and Hyperparameter Complexity: The solution requires specialized algorithms (P-Adam, ϕ-SAGA) and a novel sampling heuristic (θ-IS), which significantly increases the implementation burden and the number of hyperparameters that need careful tuning.

**Questions:**

I think this paper would win a lot if one had a better idea about the performance of the underlying JTFS algorithm in the context of sound matching. Listening to the examples, I had the feeling that the results are particularly bad for HH (higher frequencies) and noise. Some sounds are perceptually very close, and others have hardly anything in common. I had a HH, which was transformed into a chirp.

---

> ### Author Response · Authors · 2025-11-20
> **Author Response 1**
>
> Thank you for the feedback, we address each comment below. We will post another comment once the updated paper draft with the incorporated feedback has been uploaded.
>
> *Limited Experimental Scope and Generalizability: The evaluation is performed on a relatively limited number of sound matching examples, which does not permit concluding on the general applicability of the method across different audio domains or tasks.*
>
> Due to time constraints, we have not tested SCRAPL on more audio tasks. However, any generative audio task that contains spectrotemporal modulations or both transients and sustained components like music, speech, and animal vocalizations would be well suited for evaluation using SCRAPL and the JTFS. While we do not foresee any challenges extending it to other audio tasks like speech enhancement and automatic mixing, we have clarified in the paper that the SCRAPL algorithm is a general algorithm that applies to all scattering transforms with depth-first search and we suggest additional audio and vision tasks as potential directions for future work.
>
> *When listening to the audio examples, one should note that while the sound matching algorithm converges, the perceptual similarity of the original and generated sounds is not generally satisfying. In quite a few of the examples, the output sound is perceptually very far from the target. This calls the usefulness of the JTFS, and with it the SCRAPL algorithm, into question.
> I think this paper would win a lot if one had a better idea about the performance of the underlying JTFS algorithm in the context of sound matching. Listening to the examples, I had the feeling that the results are particularly bad for HH (higher frequencies) and noise. Some sounds are perceptually very close, and others have hardly anything in common. I had a HH, which was transformed into a chirp.*
>
> We agree that some of the output sounds are perceptually far from the target. The JTFS has been shown [1] to be a meaningful distance for our perceptual audio assessment experiments and we believe this issue can be attributed to limitations of the incomplete TR-808 differentiable synth implementation [2] used which is built for generating snare samples, not bass, tom, or hi-hat samples. As a result, the synth is simply unable to perfectly recreate some of the real-world analog samples in the dataset. We consider the main contribution of this work the general SCRAPL algorithm for scattering transforms and the new type of optimization tasks it enables. Consequently, we did not expend a lot of effort on customizing the decoder synth to be ideal for a specific sound matching task, nor did we do any hyperparameter exploration. We have clarified this in the paper. We also emphasize that the usefulness of the JTFS is further highlighted in experiment 1, where all other methods including SOTA fail to converge at all. In response to this feedback, we have provided additional context in the paper what advantages the JTFS provides over traditional loss functions for our perceptual quality assessment experiments.
>
> [1] Lostanlen, V., El-Hajj, C., Rossignol, M., Lafay, G., Andén, J., & Lagrange, M. (2021). Time-Frequency Scattering Accurately Models Auditory Similarities Between Instrumental Playing Techniques. EURASIP Journal on Audio, Speech, and Music Processing, 1(1), 1–21. doi:10.1186/s13636-020-00187-z
>
> [2] Shier, J., Saitis, C., Robertson, A., & McPherson, A. (2024). Real-time Timbre Remapping with Differentiable DSP. In Proceedings of the International Conference on New Interfaces for Musical Expression (pp. 377-385). arXiv:2407.04547
>
> *Increased Implementation and Hyperparameter Complexity: The solution requires specialized algorithms (P-Adam, ϕ-SAGA) and a novel sampling heuristic (θ-IS), which significantly increases the implementation burden and the number of hyperparameters that need careful tuning.*
>
> While P-Adam, P-SAGA, and $\theta$-IS indeed increase the complexity of SCRAPL, P-Adam and P-SAGA introduce no additional hyperparameters over “vanilla” Adam. $\theta$-IS introduces two new hyperparameters: how many data samples to use and the maximum number of deflated power iterations. Both did not require any tuning for all our experiments and were chosen heuristically based on available computing power. We have clarified this in the paper. Additionally, we emphasize that even without these three extra algorithms, stochastic approximation of scattering transforms (“vanilla” SCRAPL) still enables a new class of problems to be optimized and outperforms all other methods as shown in Table 2 and Figure 2.

---

> > ### Author Response · Authors · 2025-12-03
> > **Rebuttal Revision**
> >
> > We have uploaded the rebuttal revision of the paper, which incorporates the feedback from all four reviewers.

---

### Meta-Review · Area_Chair_9Y4n · 2026-01-03

**Summary:**

Followings are the main concerns pointed out by the reviewers:
* (Limited Scope/Generalizability) Several reviewers concern about the the generalizability of the method given that the experiments are done in limited scopes. For example, Reviewer HNRU points out that the experiments are done with limited number of examples and Reviewer 7Spt points out that the experiments are done mainly on audio applications.
* (Justification of the additional complexity) Several reviewers point out that the proposed methods adds additional complexity in terms of computation and hyper-parameters (Reviewer HNRU and 7Spt).
* (Convergence) Reviewer 3oJ4 concerns about the lack of convergence analysis especially comparing to the baseline JTFS.

**Reviewer Concerns:**

* (Limited Scope/generalization) This is partially addressed by authors pointing out that the method doesn't make specific assumption about the applications, thus it can potentially be applied to other applications not shown.
* (Justification of the additional complexity) This is partially addressed as the authors pointed out that the proposed method is not sensitive to some of the additional parameter.
* (Convergence) I believe the authors partially addressed the reviewer's concern through some in depth discussion. The authors pointed out the baseline JTFS is more computationally expensive than the proposed method. They also point out that the Appendix contains some proof showing the connection between SCRAPL and the full-tree ST method.

**Reviewer Scores:**

Given the authors' response and enough time for the discussion, I believe reviewer HNRU and 7Spt might likely to keep the score and Reviewer 3oJ4 might find his/her concerns partially addressed by the authors' response.

---

### Decision · Program_Chairs · 2026-01-26

Accept (Poster)